# LEARN-TO-DISTANCE: DISTANCE LEARNING FOR DETECTING LLM-GENERATED TEXT

**Hongyi Zhou**[*][1], **Jin Zhu**[*][2], **Kai Ye**[3], **Ying Yang**[1], **Erhan Xu**[†][3], **Chengchun Shi**[†][3]
[1]Tsinghua University, [2]University of Birmingham
[3]London School of Economics and Political Science

## ABSTRACT

Modern large language models (LLMs) such as GPT, Claude, and Gemini have transformed the way we learn, work, and communicate. Yet, their ability to produce highly human-like text raises serious concerns about misinformation and academic integrity, making it an urgent need for reliable algorithms to detect LLM-generated content. In this paper, we start by presenting a geometric approach to demystify rewrite-based detection algorithms, revealing their underlying rationale and demonstrating their generalization ability. Building on this insight, we introduce a novel rewrite-based detection algorithm that adaptively learns the distance between the original and rewritten text. Theoretically, we demonstrate that employing an adaptively learned distance function is more effective for detection than using a fixed distance. Empirically, we conduct extensive experiments with over 100 settings, and find that our approach demonstrates superior performance over baseline algorithms in the majority of scenarios. In particular, it achieves relative improvements from 54.3% to 75.4% over the strongest baseline across different target LLMs (e.g., GPT, Claude, and Gemini). A python implementation of our proposal is publicly available at `https://github.com/Mamba413/L2D`.

## 1 INTRODUCTION

The past few years have witnessed the emergence and rapid development of large language models (LLMs) such as GPT (Hurst et al., 2024), DeepSeek (Liu et al., 2024), Claude (Anthropic, 2024), Gemini (Comanici et al., 2025), Grok (xAI, 2025) and Qwen (Yang et al., 2025). Their impact is everywhere, from education, academia and software development to healthcare and everyday life (Arora & Arora, 2023; Chan & Hu, 2023; Hou et al., 2024). On one side of the coin, LLMs can support users with conversational question answering, help students learn more effectively, draft emails, write computer code, prepare presentation slides and more. On the other side, their ability to closely mimic human-written text also raises serious concerns, including the generation of biased or harmful content, the spread of misinformation in the news ecosystem, and the challenges related to authorship attribution and intellectual property (Dave et al., 2023; Fang et al., 2024; Messeri & Crockett, 2024; Mahajan et al., 2025; Laurito et al., 2025).

Addressing these concerns requires effective algorithms to distinguish between human-written and LLM-generated text, which has become an active and popular research direction in recent literature (see Crothers et al., 2023; Wu et al., 2025, for reviews). Existing works either *actively* detect LLM-generated text, by embedding watermarks into LLM-generated text during the design of the model (see e.g., Aaronson & Kirchner, 2023; Christ et al., 2024; Dathathri et al., 2024; Giboulot & Furon, 2024; Wouters, 2024; Wu et al., 2024; Golowich & Moitra, 2024; Li et al., 2025), or *passively*, without any prior knowledge of the watermarking process. This paper focuses on the latter category of passive detection algorithms. We review these algorithms below.

---

[*]Hongyi Zhou and Jin Zhu contributed equally to this paper and are listed in alphabetical order.
[†]Corresponding author: `e.xu2@lse.ac.uk`, `c.shi.7@lse.ac.uk`

## 1.1 RELATED WORKS

Most existing passive detection algorithms fall into the following two categories: (i) zero-shot methods and (ii) machine learning (ML)-based approaches, depending on whether they rely on external data for training the detector. Within each category, methods can be further classified into three subtypes: (1) logits-based; (2) rewrite-based, and (3) other approaches. This yields a total of 6 combinations.

**Zero-shot detection**. Zero-shot methods use only the observed text and a surrogate LLM for detection, without utilizing any additional dataset for training. They compute a statistical measure from the observed text to determine whether it was authored by a human or an LLM. The underlying rationale is that human-written text tends to produce statistics that differ (either larger or smaller) from those of LLM-generated text, and this difference can be exploited for detection (Gehrmann et al., 2019). Based on the type of statistical measure employed, these methods can be further categorized into three subtypes:

1. *Logits-based* methods construct the statistic using the logits of tokens computed by the surrogate LLM across the observed text (see e.g., Mitchell et al., 2023; Su et al., 2023; Bao et al., 2024; Hans et al., 2024; Xu et al., 2025).

2. *Rewrite-based* methods define the statistic as a suitable distance between the observed text and its rewritten (or regenerated) version (Zhu et al., 2023; Nguyen-Son et al., 2024; Yang et al., 2024; Sun & Lv, 2025). A similar principle has also been adopted for detecting machine-generated images (Wang et al., 2023; Qi et al., 2026).

3. Beyond logits or rewrite-based distances, *other* statistics have been introduced, including the intrinsic dimensionality of the observed text (Tulchinskii et al., 2023), its latent representation patterns (Chen et al., 2025b), N-gram distributions (Solaiman et al., 2019) and maximum mean discrepancy (Zhang et al., 2024; Song et al., 2025).

**ML-based detection**. ML-based methods leverage external human- and LLM-authored text to enhance the detection power of zero-shot methods. A primary approach is to formulate the detection task as a classification problem and utilize external data to train the classifier. Similar to zero-shot methods, ML-based approaches can also be categorized into three subtypes:

1. *Logits-based* methods fine-tune the surrogate LLM's logits to improve the classification accuracy. Various LLMs have been employed in the literature, including RoBERTa (Solaiman et al., 2019; Guo et al., 2023), BERT (Ippolito et al., 2020), DistilBERT (Mitrović et al., 2023), and reward models for aligning LLMs with human feedback (Lee et al., 2024). Recent works have extended these methods to more challenging scenarios, including handling adversarial attacks (Hu et al., 2023; Koike et al., 2024; Sadasivan et al., 2025), short texts such as tweets and reviews (Tian et al., 2024), black-box settings under diverse prompts (Zeng et al., 2024; Chen et al., 2025a), and accommodating statistical inference (Zhou et al., 2026).

2. *Rewrite-based* methods either use the distance between the observed text and its rewritten version as an input feature for training the classifier (Mao et al., 2024; Yu et al., 2024b; Huang et al., 2025; Park et al., 2025), or apply ML to fine-tune the the rewriting model itself to improve the detection accuracy (Hao et al., 2025).

3. *Other* methods extract features beyond logits or rewrite-based distances, and then apply ML algorithms to these features for classification. Examples of features being utilized range from classical N-grams and term frequency–inverse document frequency widely used in natural language processing (Solaiman et al., 2019), to more complex representations such as various combinations of features constructed based on token probabilities (Verma et al., 2024), cross-entropy loss between the text and a surrogate LLM (Guo et al., 2024a), hidden latent representations (Yu et al., 2024a) and features learned via multi-level contrastive learning (Guo et al., 2024b), and even classification probabilities of fine-tuned LLMs (Abburi et al., 2023).

## 1.2 CONTRIBUTIONS

Our proposal falls under the category of ML-based, rewrite-based detection. We study a commonly encountered setting in practice, where LLM-authored text is generated using prompts that are unobserved by the detector. Our main contributions are as follows:

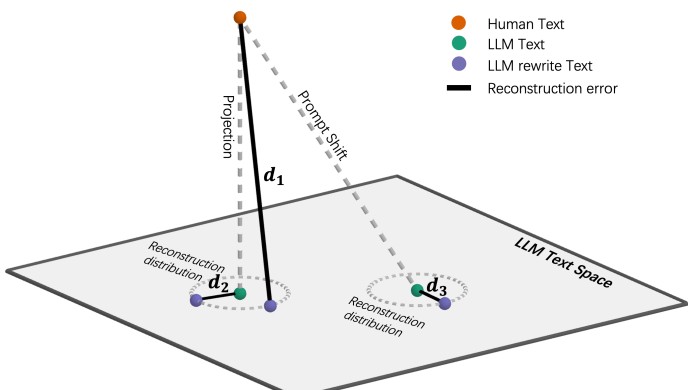

Figure 1: The rationale behind rewrite-based methods: the brown dot represents a human-authored text after embedding, while the two green dots represent its projection onto the LLM subspace and an LLM-generated text produced from an unobserved prompt, respectively. From left to right, the purple dots denote the reconstructions of the first green dot, the brown dot and the second green dot. As illustrated, $d_1 > d_2$, indicating that the reconstruction error for human text is larger than that for LLM-generated text, which aligns with Proposition 1. Additionally, $d_1 > d_3$ suggests that rewrite-based methods remain robust to prompt-induced distribution shifts, as formalized in Proposition 2.

- *Methodologically*, we develop a new rewrite-based method for detecting LLM-generated text. Unlike existing approaches that primarily employ a fixed distance to compare the original text with its rewritten version, we propose to adaptively learn this distance via ML. Our proposal better discriminates between LLM- and human-authored text (see Figure 2 for a graphical illustration), leading to substantial performance gains.

- *Theoretically*, we develop a geometric approach to demystify the rationale behind rewrite-based methods (see Figure 1 for illustration and Proposition 1 for the detailed statement). We next show that these methods generalize well to unobserved prompts (Proposition 2). Finally, we demonstrate the rationale for learning a distance function rather than relying on a fixed distance (Proposition 3).

- *Empirically*, we conduct comprehensive experiments across **24** datasets, **7** target language models, and **3** types of unseen prompts, covering over **100** settings. Our results show that: (i) our approach outperforms **12** state-of-the-art methods, achieving average relative improvements of **41.5**% to **75.4**% over the strongest baseline across different target LLMs baseline (Sections 4.1 and 4.2); (ii) our approach is more robust than existing methods under adversarial attacks (Section 4.3); (iii) learning the distance function provides substantial benefits, with an average relative improvement of **96**% over using a fixed distance (see the ablation study in Section 4.4).

## 2 REWRITE-BASED METHODS: BUILDING INTUITION

In this section, we present a geometric framework for understanding rewrite-based detection methods, revealing their underlying rationale and demonstrating their robustness to unseen prompts.

Let $X$ denote the target text under detection. We study the problem of determining whether $X$ is authored by a suspected target LLM, or by a human. Rewrite-based methods are straightforward to describe: they first prompt the target LLM to rephrase the original text and then measure the discrepancy between the original text $X$ and the LLM's reconstruction (denoted by $\mathcal{R}(X)$) under a distance metric $d$. These methods rely on the observation that, compared to human-authored text, machine-generated text should be closer to its reconstruction (Mao et al., 2024; Yang et al., 2024). In the following, we will formally prove this assertion from a geometric perspective.

**Building intuition**. We begin with some notations and hypotheses. Let $(\mathcal{X}, \mathcal{B})$ denote a measurable space of texts (after embedding).

**Assumption 1.** Assume $\mathcal{X}$ is a Hilbert space with inner product $\langle \cdot, \cdot \rangle$, induced norm $|\cdot|$, and metric $d^*(x, y) := |x - y|$ for any $x, y \in \mathcal{X}$.

This assumption is reasonable since texts are typically mapped into a vector space where each token is represented by a scalar (Mikolov et al., 2013), and padding is commonly applied to ensure all texts share the same dimensionality.

Let $\mathcal{H}$ and $\mathcal{M}$ denote the subspaces corresponding to texts authored by humans and the target LLM, respectively. We use $p$ and $q$ to represent their respective probability distributions. We also define the projection operator $\Pi$ onto $\mathcal{M}$,

$$\Pi_{\mathcal{M}}(x) \;=\; \arg\min_{y \in \mathcal{M}} d^*(x, y), \tag{1}$$

which projects a given text $x \in \mathcal{X}$ to its closest point in $\mathcal{M}$, produced by the target LLM.

**Assumption 2.** $q$ is the projection of $p$ under $\Pi_{\mathcal{M}}$, i.e., if $\boldsymbol{X} \sim p$ then $\Pi_{\mathcal{M}}(\boldsymbol{X}) \sim q$.

Assumption 2 is our key hypothesis, which reflects the geometric relationship between human- and LLM-authored text. Intuitively, it implies that all LLM-generated texts can be viewed as a projection of human-written text onto a specific subspace. This assumption is reasonable because (i) LLMs are trained on massive corpora of human-authored text with the objective of approximating the distribution of human language; (ii) LLM's output space is constrained by the model's architecture and learned parameters, and is thus different from the human text space. Therefore, the mapping from human text to LLM-generated text can be interpreted as a projection: a transformation that preserves semantic meanings while restricting outputs to the region defined by the model.

**Assumption 3.** For any human-written text $x \in \mathcal{H}$, $\mathcal{R}(x)$ has the same probability distribution function to $\mathcal{R}(\Pi_{\mathcal{M}}(x))$.

Here, for a fixed text $x$, we allow its reconstruction $\mathcal{R}(x)$ to be random. This is because LLM outputs are typically stochastic due to the use of a nonzero temperature during inference. Assumption 3 essentially requires the reconstructions of a human-written text $x$ and its projection $\Pi_{\mathcal{M}}(x)$ to share the same distribution. This holds when the reconstruction can be written as

$$\mathcal{R}(x) = \Pi_{\mathcal{M}}(x) + e, \tag{2}$$

for some random error $e$ that lies on the space of $\mathcal{M}$. Equation 2 suggests that the rewriting process can be viewed as a two-step procedure: first, the input text is projected onto the LLM subspace, and then a small perturbation $e$ is added to the projected text, while preserving the projected text's semantic meaning.

**Proposition 1.** *Under Assumptions 1, 2 and 3, we have*

$$\mathbb{E}_{\boldsymbol{X} \sim p}\big[d^*(\boldsymbol{X}, \mathcal{R}(\boldsymbol{X}))\big] \geq \mathbb{E}_{\boldsymbol{X} \sim q}\big[d^*(\boldsymbol{X}, \mathcal{R}(\boldsymbol{X}))\big],$$

*with equality if and only if $p$ is supported on $\mathcal{M}$.*

Proposition 1 formally establishes the validity of rewrite-based methods, and proves that human-written text's reconstruction error (the distance between a text and its reconstruction) is on average larger than that of LLM-generated text. The equality holds only under the idealized scenario where the LLM's output space perfectly replicates the human text space.

Intuitively, this result follows because reconstructions always lie within the LLM subspace $\mathcal{M}$, whereas human-authored text may lie farther away from $\mathcal{M}$. Figure 1 provides a graphical illustration: the reconstruction error for human text ($d_1$) is clearly larger than that for LLM-generated text ($d_2$).

**Generalization to unseen prompts**. In practice, LLM-generated text is often produced under a variety of writing prompts (e.g., "polish this paragraph" or "help me rephrase"). The presence of such prompts induces a distributional shift: the resulting LLM-generated text no longer follows the original distribution $q$, but instead depends on the specific prompt, which we denote by $q_{\text{prompt}}$. This shift is illustrated in Figure 1, where the prompt alters the location of the generated text in the embedding space.

Rewrite-based methods can generalize effectively to such shifts, provided that the perturbation $e$ in equation 2 does not substantially distort the semantic meaning of $\Pi_{\mathcal{M}}(x)$. We formalize this intuition in the following proposition.

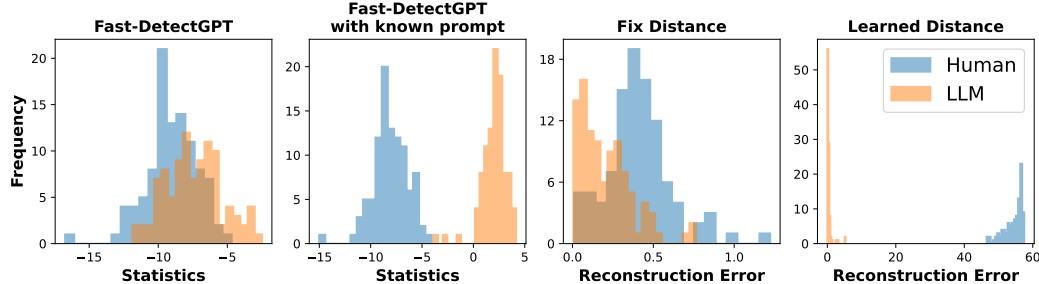

Figure 2: Histograms comparing the statistics constructed by Fast-DetectGPT (a state-of-the-art logits-based detector) and the reconstruction errors of rewrite-based methods between human-written and LLM-rewritten news text. The first two panels show that Fast-DetectGPT effectively distinguishes human- from LLM-authored text only when the prompt to produce LLM-generated text is known. The last two panels show that the proposed learned distance provides a much clearer separation than using a fixed distance.

**Proposition 2.** *Assume equation 2 holds. Let $\epsilon > 0$ denote some positive constant such that $|e| \leq \epsilon$ almost surely. Then under Assumption 1, we have*

$$\mathbb{E}_{\boldsymbol{X} \sim p}\big[d^*(\boldsymbol{X}, \mathcal{R}(\boldsymbol{X}))\big] - \mathbb{E}_{\boldsymbol{X} \sim q_{prompt}}\big[d^*(\boldsymbol{X}, \mathcal{R}(\boldsymbol{X}))\big] \geq \mathbb{E}_{\boldsymbol{X} \sim p}|\boldsymbol{X} - \Pi_{\mathcal{M}}(\boldsymbol{X})| - O(\epsilon).$$

Proposition 2 provides a lower bound to quantify the difference in reconstruction error between human- and LLM-authored text. The bound depends on two factors: (i) the average gap between human and LLM-generated text, characterized by the norm of the projection $\mathbb{E}_{\boldsymbol{X} \sim p}|\boldsymbol{X} - \Pi_{\mathcal{M}}(\boldsymbol{X})|$; (ii) the magnitude of the perturbation $e$.

Figure 1 offers a graphical illustration: despite the shift introduced by the prompt, as long as $e$ remains small, the reconstruction error for human text ($d_1$) can still be substantially larger than that for LLM-generated text ($d_3$). In practice, minimizing $e$ requires careful design of the rewriting prompt to preserve the input text's semantic meaning. This can be achieved through prompt engineering or by adaptively learning the rewrite model (Hao et al., 2025).

## 3 ADAPTIVE DISTANCE LEARNING

**Limitations of existing approaches**. We begin by discussing the limitations of existing logits-based and rewrite-based detection methods to better motivate our proposed approach:

- Logit-based methods, such as DetectGPT (Mitchell et al., 2023) and Fast-DetectGPT (Bao et al., 2024), construct the detection statistics using the log-probability $\log q(x)$ of the text. However, their performance tends to degrade when the text is generated under unseen prompts (see the first two panels of Figure 2 for illustration). This arises because the true conditional distribution $\log q(x \mid \text{prompt})$ differs from the marginal distribution $\log q(x)$ used by the detector, leading to the misspecification of the detection statistic.

- The effectiveness of rewrite-based methods relies on choosing an appropriate distance function to distinguish human- from LLM-authored text, and the optimal distance function may differ largely from standard Euclidean distance due to the complex geometry of text embeddings. Nonetheless, existing rewrite-based methods often use fixed, hand-crafted distance, such as N-gram-based distance (Yang et al., 2024), Levenshtein distance (Mao et al., 2024), and negative BERTScore or BARTScore (Zhang et al., 2019; Yuan et al., 2021), which may not generalize well across target language models, datasets or unobserved prompts.

To elaborate on the second point, we provide a proposition below to mathematically characterize the form of the optimal distance function.

**Proposition 3.** *Consider the class of distance functions $d$ whose range is bounded between $0$ to and some positive constant $M > 0$. Within this function class, and under mild regularity conditions (see Appendix A), any distance function $d_{opt}$ satisfying*

$$d_{opt}(\boldsymbol{X}, \boldsymbol{Y}) = \begin{cases} 0, & \text{if both } \boldsymbol{X} \text{ and } \boldsymbol{Y} \in \mathcal{M}; \\ M, & \text{if one of } \boldsymbol{X} \text{ or } \boldsymbol{Y} \in \mathcal{M} \text{ and the other} \in \mathcal{H} \cap \mathcal{M}^c, \end{cases}$$

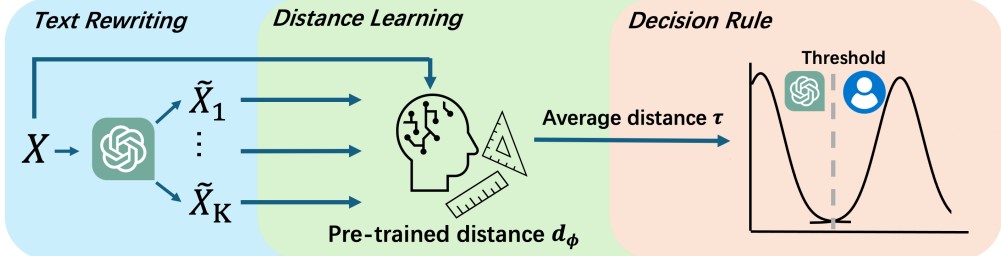

Figure 3: Workflow of the proposal. Our method adaptively learn a distance metric to measure the discrepancy between human and LLM-generated texts for detection.

*maximizes the gap in the reconstruction error*

$$\mathbb{E}_{\boldsymbol{X} \sim p}\big[d(\boldsymbol{X}, \mathcal{R}(\boldsymbol{X}))\big] - \mathbb{E}_{\boldsymbol{X} \sim q_{prompt}}\big[d(\boldsymbol{X}, \mathcal{R}(\boldsymbol{X}))\big].$$

Proposition 3 shows that the optimal distance function should assign the smallest possible distance (zero) when both the input and rewritten text are generated with the LLM, and the largest distance $M$ when one is LLM-generated and the other is human-written. Crucially, this optimal distance depends on the target LLM to be detected, since different LLMs induce different generative subspaces $\mathcal{M}$. However, existing rewrite-based detectors rely entirely on fixed distance functions (e.g., editing distance, embedding similarity). As a result, a distance that works well for one model may perform poorly with another, limiting their ability to generalize across different LLMs.

**Our proposal**. Motivated by the aforementioned limitations, we adopt the rewrite-based approach, and propose to adaptively learn the distance function to improve the detection performance. More specifically, assume we have access to a human-authored corpus $\mathcal{D}_h$ and an LLM-generated corpus $\mathcal{D}_m$, both of which are readily available in practice. For instance, $\mathcal{D}_h$ can be obtained by web-scraping Wikipedia, while $\mathcal{D}_m$ can be constructed by prompting the target LLM (e.g., GPT, Gemini, or Grok). We next learn the distance function $d$, parameterized by some parameter $\phi$, that maximizes the discrepancy between the reconstructions errors:

$$\mathbb{E}_{\boldsymbol{X} \sim D_h}\big[d(\boldsymbol{X}, \mathcal{R}(\boldsymbol{X}))\big] - \mathbb{E}_{\boldsymbol{X} \sim D_m}\big[d(\boldsymbol{X}, \mathcal{R}(\boldsymbol{X}))\big].$$

In our implementation, we parameterize the distance function via

$$d_\phi(\boldsymbol{X}_1, \boldsymbol{X}_2) = \left| \frac{\log p_\phi(\boldsymbol{X}_1)}{\texttt{len}(\boldsymbol{X}_1)} - \frac{\log p_\phi(\boldsymbol{X}_2)}{\texttt{len}(\boldsymbol{X}_2)} \right|, \tag{3}$$

where $p_\phi$ is a language model parameterized by $\phi$ and $\texttt{len}(\cdot)$ computes the number of tokens of the input text. It is straightforward to show that $d_\phi$ in equation 3 satisfies the property of a (pseudo)-distance: (i) It is non-negative; (ii) It equals zero whenever $\boldsymbol{X}_1 = \boldsymbol{X}_2$; (iii) It satisfies the triangle inequality.

Our choice of equation 3 is also motivated by the form of the optimal distance function $d_{\text{opt}}$ in Proposition 3. It can be viewed as a soft relaxation of $d_{\text{opt}}$ which is binary and involves hard indicators, making the objective function continuous and the optimization tractable. Notably, when $p_\phi$ assigns any $\boldsymbol{X} \in \mathcal{M}$ a probability proportional to $\kappa^{\text{len}(\boldsymbol{X})}$ for some $0 < \kappa < 1$, the distance between any two texts produced by the LLM will be exactly zero. To the contrary, when $p_\phi$ assigns very low probability to human-written text, the resulting distance between human- and LLM-authored text will be large.

Our above discussion also highlights the need to adaptively learn the language model $p_\phi$ as opposed to using a fixed model. The ideal $p_\phi$ should: (i) assign low probability to human-authored text; (ii) assign probability more uniformly across tokens for LLM-generated text. This differs from conventional LLMs, which aim to produce coherent, human-like text and therefore tend to assign high probability to human-authored text. Empirically, as demonstrated in the last two panels of Figure 2, the learned distance more effectively distinguishes between human- and LLM-authored text compared to a fixed distance. Our experiments in Section 4.4 also show that, the learned distance function yields substantial improvements over using the initial pre-trained LLM.

To solve the optimization, we initialize $p_\phi$ with a pre-trained LLM and fine-tune a small subset of its parameters to facilitate the computation. This can be done by updating only the final layer

or employing low-rank adaptation (LoRA, Hu et al., 2022). Furthermore, since the rewritten text $\mathcal{R}(\boldsymbol{X})$ is stochastic, we mitigate its randomness by generating multiple reconstructions. Given a text $\boldsymbol{X}$, we obtain $K$ reconstructions $\widetilde{\boldsymbol{X}}_1, \ldots, \widetilde{\boldsymbol{X}}_K$, and estimate the reconstruction error as the average: $K^{-1} \sum_{k=1}^{K} d(\boldsymbol{X}, \widetilde{\boldsymbol{X}}_k)$. We classify $\boldsymbol{X}$ as LLM-generated if this value is smaller than a predetermined threshold, and as human-authored otherwise. We summarize our procedure in Figure 3.

## 4 EXPERIMENTS

We conduct extensive experiments to evaluate the effectiveness of our approach. To save space, we defer additional implementation details to Appendix D. Our empirical study is designed to answer the following three questions:

1. *How does our method perform compared to state-of-the-art approaches under different prompts?*
2. *How robust is our method under adversarial attacks?*
3. *To what extent does learning the distance improve the detection accuracy?*

To answer the first question, we compare our method, denoted as L2D (short for learn-to-distance), against **12** representative baseline detectors in Sections 4.1 and 4.2, covering both zero-shot (left) and ML-based methods (right):

- *Likelihood* (Gehrmann et al., 2019)
- Intrinsic dimension estimation (*IDE*, Tulchinskii et al., 2023)
- Log rank ratio (*LRR*, Su et al., 2023)
- Fast-DetectGPT (*FDGPT*, Bao et al., 2024)
- *BARTScore* (Zhu et al., 2023)
- *Binoculars* (Hans et al., 2024)

- *RoBERTa* (Solaiman et al., 2019)
- *RADAR* (Hu et al., 2023)
- *RADIAR* (Mao et al., 2024)
- AdaDetectGPT (*ADGPT*, Zhou et al., 2025)
- Imitate before detection (*ImBD*, Chen et al., 2025a)
- Learning to rewriting (*L2R*, Hao et al., 2025)

We also employ **24** datasets and consider **6** commonly used target LLMs such as Llama-3-70B-Instruct (Dubey et al., 2024), Claude-3.5, GPT series (GPT-3.5 Turbo and GPT-4o, OpenAI, 2022; Hurst et al., 2024), and Gemini models (Gemini 1.5 Pro and Gemini 2.5 Flash, Team et al., 2024; Comanici et al., 2025) for generating LLM-written text.

To answer the second and third questions, we further consider settings under paraphrasing and decoherence attacks in Section 4.3 and compare against a variant of our approach that uses the initial pre-trained model $p_\phi$ without fine-tuning as the distance function in Section 4.4.

Throughout, we have taken care to ensure fairness in all experimental comparisons. Specifically: (i) Both the baseline methods and our algorithm use the same base model, `google/gemma-2-9b-it`, as the rewrite and/or scoring model to maintain consistency. (ii) For each input text, we use the same set of rewritten texts across all rewrite-based algorithms to ensure a fair comparison. (iii) For algorithms such as ImBD that involve fine-tuning, we use the same optimization hyperparameters (e.g., number of epochs, learning rate) as ours across all cases to ensure fairness in training.

Finally, the area under the curve (AUC) is used as the metric for evaluation.

### 4.1 EXPERIMENTS ON DIVERSE DATASETS

We first evaluate our method on the dataset released by Hao et al. (2025)[1], which consists of human-written text from **21** domains, including academic writing, business, code, sports and religion. For each human-written sample, four LLM-generated versions were created using Llama-3-70B-Instruct, Gemini 1.5 Pro, GPT-3.5 Turbo and GPT-4o, respectively, yielding a total of **84** settings. Refer to Hao et al. (2025) for the detailed prompts used to produce these LLM-generated texts.

---

[1]`https://github.com/ranhli/l2r_data`

Table 1: AUC scores of various detectors for detecting text generated by GPT-3.5 Turbo. The highest scores are highlighted in cyan, the second best in orange. The last two columns show the percentage absolute gain (AG) and relative gain (RG) over the best baseline. With baseline score $x$ and our score $y$, the absolute gain is $(y-x) \times 100\%$, and the relative gain is $(y-x)/(1-x) \times 100\%$.

| Dataset | Likelihood | LRR | IDE | BARTScore | FDGPT | Binoculars | RoBERTa | RADAR | ADGPT | RAIDAR | ImBD | L2D | AG (%) | RG (%) |
|---|---|---|---|---|---|---|---|---|---|---|---|---|---|---|
| AcademicResearch | 0.582 | 0.557 | 0.571 | 0.561 | 0.542 | 0.532 | 0.510 | 0.718 | 0.544 | 0.812 | 0.919 | 0.948 | 2.915 | 35.8 |
| ArtCulture | 0.529 | 0.539 | 0.508 | 0.620 | 0.556 | 0.580 | 0.605 | 0.618 | 0.549 | 0.618 | 0.732 | 0.835 | 10.285 | 38.4 |
| Business | 0.532 | 0.563 | 0.574 | 0.639 | 0.657 | 0.656 | 0.564 | 0.587 | 0.518 | 0.704 | 0.861 | 0.914 | 5.314 | 38.1 |
| Code | 0.677 | 0.530 | 0.601 | 0.551 | 0.556 | 0.568 | 0.525 | 0.702 | 0.575 | 0.539 | 0.771 | 0.906 | 13.443 | 58.8 |
| EducationMaterial | 0.561 | 0.813 | 0.705 | 0.808 | 0.785 | 0.707 | 0.708 | 0.847 | 0.557 | 0.961 | 0.996 | 0.973 | — | — |
| Entertainment | 0.601 | 0.645 | 0.725 | 0.866 | 0.805 | 0.745 | 0.750 | 0.887 | 0.510 | 0.875 | 0.983 | 0.982 | — | — |
| Environmental | 0.672 | 0.636 | 0.608 | 0.854 | 0.830 | 0.770 | 0.680 | 0.647 | 0.569 | 0.850 | 0.932 | 0.984 | 5.201 | 76.7 |
| Finance | 0.546 | 0.608 | 0.618 | 0.819 | 0.730 | 0.699 | 0.678 | 0.647 | 0.507 | 0.750 | 0.956 | 0.987 | 3.086 | 69.6 |
| FoodCusine | 0.569 | 0.534 | 0.524 | 0.739 | 0.639 | 0.625 | 0.562 | 0.526 | 0.569 | 0.735 | 0.869 | 0.969 | 10.072 | 76.7 |
| GovernmentPublic | 0.530 | 0.551 | 0.572 | 0.680 | 0.697 | 0.692 | 0.612 | 0.639 | 0.531 | 0.748 | 0.903 | 0.923 | 1.951 | 20.1 |
| LegalDocument | 0.740 | 0.509 | 0.807 | 0.637 | 0.741 | 0.701 | 0.596 | 0.819 | 0.503 | 0.595 | 0.991 | 0.994 | 0.250 | 29.2 |
| LiteratureCreativeWriting | 0.541 | 0.520 | 0.705 | 0.645 | 0.634 | 0.550 | 0.637 | 0.866 | 0.653 | 0.784 | 0.993 | 0.996 | 0.316 | 45.9 |
| MedicalText | 0.553 | 0.564 | 0.538 | 0.591 | 0.620 | 0.600 | 0.519 | 0.629 | 0.556 | 0.654 | 0.754 | 0.828 | 7.374 | 29.9 |
| NewsArticle | 0.655 | 0.674 | 0.656 | 0.555 | 0.513 | 0.506 | 0.626 | 0.861 | 0.616 | 0.785 | 0.893 | 0.968 | 7.488 | 70.0 |
| OnlineContent | 0.539 | 0.525 | 0.512 | 0.711 | 0.654 | 0.632 | 0.596 | 0.604 | 0.541 | 0.743 | 0.844 | 0.950 | 10.630 | 68.2 |
| PersonalCommunication | 0.555 | 0.521 | 0.515 | 0.602 | 0.541 | 0.547 | 0.526 | 0.581 | 0.555 | 0.653 | 0.755 | 0.922 | 16.660 | 68.0 |
| ProductReview | 0.625 | 0.628 | 0.553 | 0.803 | 0.688 | 0.675 | 0.611 | 0.591 | 0.529 | 0.728 | 0.880 | 0.971 | 9.107 | 75.7 |
| Religious | 0.741 | 0.642 | 0.662 | 0.884 | 0.534 | 0.543 | 0.579 | 0.869 | 0.648 | 0.812 | 0.970 | 0.957 | — | — |
| Sports | 0.511 | 0.531 | 0.510 | 0.522 | 0.584 | 0.592 | 0.561 | 0.606 | 0.527 | 0.664 | 0.821 | 0.910 | 8.883 | 49.6 |
| TechnicalWriting | 0.594 | 0.559 | 0.569 | 0.594 | 0.555 | 0.537 | 0.516 | 0.739 | 0.519 | 0.818 | 0.944 | 0.994 | 5.020 | 89.4 |
| TravelTourism | 0.590 | 0.538 | 0.571 | 0.600 | 0.550 | 0.525 | 0.531 | 0.741 | 0.503 | 0.824 | 0.917 | 0.989 | 7.243 | 87.0 |
| Average | 0.593 | 0.580 | 0.600 | 0.680 | 0.639 | 0.618 | 0.595 | 0.701 | 0.551 | 0.745 | 0.890 | 0.948 | 5.789 | 52.5 |
| Std | 0.066 | 0.071 | 0.080 | 0.113 | 0.095 | 0.078 | 0.066 | 0.112 | 0.042 | 0.099 | 0.082 | 0.047 | — | — |

Results are reported in Tables 1, B1 and Tables B2 – B5 in Appendix B. It can be seen that our method achieves the best performance across nearly all combinations of datasets and target models. We focus on comparison against four baselines: (i) FDGPT, a training-free, logits-based zero-shot approach; (ii) ADGPT and (iii) ImBD, both ML-based variants of FDGPT. We include them because, similar to our algorithm, these methods require training. Note that ImBD typically ranks second overall and is the strongest among logits-based approaches; (iv) L2R, a rewrite-based method that also employs ML but learns the rewrite model rather than the distance function. We make two observations:

1. First, our approach consistently achieves substantially larger AUC scores than FDGPT. Notice that, in Tables 1, B1 and B3, the training and testing data differ in terms of models or data contexts, which reduces the inherent advantage of ML-based approaches over zero-shot methods such as FDGPT. Even under these shifts, our method continues to achieve the best performance in most cases. *This comparison highlights our algorithm's robustness to distributional shifts between the training and testing data, as well as its effectiveness relative to zero-shot methods.*

2. Second, as shown in Tables 1, B1, B2 and B3, our approach outperforms ImBD on most datasets (16 – 19 out of 21), and the relative gain can reach up to 89.4% (see the rightmost column). *This comparison highlights the advantage of rewrite-based methods over logits-based methods.*

3. Third, since L2R does not provide public code, we directly compare against the reported results in their paper. Table B5 shows that our method outperforms L2R on 20 out of 21 datasets, and often by a large margin. *This comparison suggests that, compared with learning to rewrite, learning a distance function is more effective for rewrite-based detection.*

## 4.2 EXPERIMENTS UNDER DIFFERENT PROMPTS

Next, following Chen et al. (2025a), we examine **three** scenarios that use different types of unseen prompts to generate LLM text: (i) *rewrite*, where the LLM rewrites a human-authored text while preserving its semantic meaning; (ii) *expand*, where the LLM elaborates on the text according to a style randomly selected from various options (e.g., formal, literary); and (iii) *polish*, where the LLM refines the text based on the randomly chosen style.

We also consider **three** widely used benchmark datasets (Bao et al., 2024; Chen et al., 2025a): (i) *Wiki*, which consists of Wikipedia-style question answering data (Rajpurkar et al., 2016); (ii) *Story*, which focuses on story generation (Fan et al., 2018); and (iii) *News*, which is concerned with news summarization (Narayan et al., 2018).

We further generate LLM-authored text using **three** recent and popular proprietary models: (i) *GPT-4o*; (ii) *Claude-3.5-Haiku* and (iii) *Gemini-2.5-Flash*. This yields a total of **27** settings. Details on how these texts were generated are provided in Appendix D.

Table 2: AUC scores across datasets, models, and tasks; best method highlighted in `cyan`, second best in `orange`. The last two rows show the absolute gain and relative gain of our approach over the best baseline in percentage. On Claude-3.5, GPT-4o, and Gemini-2.5, the average absolute gain are 3.83%, 2.57%, 0.58%, and relative gain are 75.40%, 65.74%, 54.35%

| Dataset | Method | Claude-3.5 | | | | GPT-4o | | | | Gemini | | | |
|---|---|---|---|---|---|---|---|---|---|---|---|---|---|
| | | rewrite | polish | expand | Avg. | rewrite | polish | expand | Avg. | rewrite | polish | expand | Avg. |
| News | Likelihood | 0.598 | 0.604 | 0.645 | 0.616 | 0.572 | 0.587 | 0.539 | 0.566 | 0.594 | 0.579 | 0.732 | 0.635 |
| | LRR | 0.594 | 0.626 | 0.636 | 0.619 | 0.633 | 0.620 | 0.559 | 0.604 | 0.656 | 0.601 | 0.717 | 0.658 |
| | Binoculars | 0.555 | 0.634 | 0.709 | 0.633 | 0.535 | 0.567 | 0.631 | 0.578 | 0.507 | 0.632 | 0.589 | 0.576 |
| | IDE | 0.606 | 0.686 | 0.726 | 0.673 | 0.577 | 0.736 | 0.696 | 0.670 | 0.608 | 0.672 | 0.716 | 0.665 |
| | FDGPT | 0.524 | 0.610 | 0.686 | 0.607 | 0.508 | 0.561 | 0.641 | 0.570 | 0.507 | 0.617 | 0.586 | 0.570 |
| | BARTScore | 0.728 | 0.583 | 0.563 | 0.625 | 0.653 | 0.526 | 0.549 | 0.576 | 0.567 | 0.606 | 0.671 | 0.615 |
| | RoBERTa | 0.544 | 0.524 | 0.546 | 0.538 | 0.509 | 0.532 | 0.568 | 0.536 | 0.501 | 0.566 | 0.567 | 0.545 |
| | RADAR | 0.744 | 0.805 | 0.912 | 0.821 | 0.774 | 0.966 | 0.994 | 0.911 | 0.807 | 0.858 | 0.920 | 0.862 |
| | ADGPT | 0.518 | 0.616 | 0.569 | 0.567 | 0.617 | 0.644 | 0.561 | 0.608 | 0.514 | 0.543 | 0.502 | 0.520 |
| | RAIDAR | 0.934 | 0.919 | 0.942 | 0.932 | 0.882 | 0.900 | 0.866 | 0.882 | 0.800 | 0.948 | 0.921 | 0.890 |
| | ImBD | 0.920 | 0.915 | 0.986 | 0.940 | 0.866 | 0.978 | 0.985 | 0.943 | 0.877 | 0.952 | 0.966 | 0.932 |
| | L2D | 1.000 | 0.989 | 1.000 | 0.996 | 0.994 | 1.000 | 1.000 | 0.998 | 1.000 | 1.000 | 1.000 | 1.000 |
| | *Abs. Gain (%)* | 6.6 | 7.0 | 1.4 | 5.6 | 11.2 | 2.2 | 1.5 | 5.5 | 12.3 | 4.8 | 3.4 | 6.8 |
| | *Rel. Gain (%)* | 100.0 | 86.6 | 100.0 | 94.0 | 94.7 | 100.0 | 100.0 | 96.3 | 100.0 | 100.0 | 100.0 | 100.0 |
| Wiki | Likelihood | 0.519 | 0.532 | 0.562 | 0.538 | 0.546 | 0.553 | 0.649 | 0.583 | 0.505 | 0.512 | 0.533 | 0.517 |
| | LRR | 0.532 | 0.508 | 0.540 | 0.527 | 0.541 | 0.612 | 0.695 | 0.616 | 0.522 | 0.508 | 0.536 | 0.522 |
| | Binoculars | 0.608 | 0.667 | 0.762 | 0.679 | 0.619 | 0.717 | 0.862 | 0.733 | 0.571 | 0.768 | 0.793 | 0.711 |
| | IDE | 0.565 | 0.621 | 0.613 | 0.600 | 0.584 | 0.712 | 0.682 | 0.659 | 0.573 | 0.642 | 0.699 | 0.638 |
| | FDGPT | 0.587 | 0.646 | 0.739 | 0.658 | 0.597 | 0.712 | 0.867 | 0.725 | 0.557 | 0.748 | 0.791 | 0.699 |
| | BARTScore | 0.760 | 0.634 | 0.520 | 0.638 | 0.785 | 0.592 | 0.529 | 0.635 | 0.605 | 0.590 | 0.615 | 0.603 |
| | RoBERTa | 0.635 | 0.659 | 0.759 | 0.684 | 0.565 | 0.590 | 0.522 | 0.559 | 0.638 | 0.740 | 0.782 | 0.720 |
| | RADAR | 0.533 | 0.507 | 0.620 | 0.553 | 0.541 | 0.814 | 0.933 | 0.763 | 0.550 | 0.564 | 0.680 | 0.598 |
| | ADGPT | 0.518 | 0.616 | 0.569 | 0.567 | 0.617 | 0.644 | 0.561 | 0.608 | 0.514 | 0.543 | 0.502 | 0.520 |
| | RAIDAR | 0.889 | 0.900 | 0.920 | 0.903 | 0.845 | 0.871 | 0.851 | 0.856 | 0.848 | 0.927 | 0.950 | 0.908 |
| | ImBD | 0.952 | 0.954 | 0.976 | 0.961 | 0.875 | 0.967 | 0.986 | 0.943 | 0.874 | 0.964 | 0.956 | 0.931 |
| | L2D | 0.955 | 0.942 | 0.953 | 0.950 | 0.963 | 0.987 | 0.993 | 0.981 | 0.983 | 0.982 | 0.988 | 0.984 |
| | *Abs. Gain (%)* | 0.2 | — | — | — | 8.8 | 1.9 | 0.6 | 3.8 | 10.9 | 1.8 | 3.2 | 5.3 |
| | *Rel. Gain (%)* | 5.0 | — | — | — | 70.6 | 59.1 | 45.8 | 66.4 | 86.3 | 49.6 | 72.6 | 76.9 |
| Story | Likelihood | 0.502 | 0.532 | 0.587 | 0.541 | 0.623 | 0.740 | 0.814 | 0.725 | 0.512 | 0.656 | 0.702 | 0.623 |
| | LRR | 0.556 | 0.540 | 0.596 | 0.564 | 0.570 | 0.728 | 0.739 | 0.679 | 0.504 | 0.563 | 0.632 | 0.566 |
| | Binoculars | 0.595 | 0.663 | 0.755 | 0.671 | 0.674 | 0.739 | 0.806 | 0.740 | 0.624 | 0.832 | 0.927 | 0.794 |
| | IDE | 0.616 | 0.610 | 0.632 | 0.619 | 0.575 | 0.650 | 0.673 | 0.633 | 0.580 | 0.579 | 0.609 | 0.589 |
| | FDGPT | 0.571 | 0.635 | 0.743 | 0.650 | 0.655 | 0.735 | 0.808 | 0.733 | 0.603 | 0.000 | 0.918 | 0.507 |
| | BARTScore | 0.767 | 0.706 | 0.566 | 0.680 | 0.724 | 0.754 | 0.685 | 0.721 | 0.708 | 0.733 | 0.674 | 0.705 |
| | RoBERTa | 0.588 | 0.586 | 0.660 | 0.611 | 0.540 | 0.504 | 0.539 | 0.527 | 0.571 | 0.569 | 0.657 | 0.599 |
| | RADAR | 0.597 | 0.614 | 0.510 | 0.574 | 0.507 | 0.756 | 0.827 | 0.697 | 0.560 | 0.513 | 0.619 | 0.564 |
| | ADGPT | 0.755 | 0.746 | 0.789 | 0.763 | 0.617 | 0.698 | 0.655 | 0.657 | 0.729 | 0.692 | 0.658 | 0.693 |
| | RAIDAR | 0.861 | 0.767 | 0.847 | 0.825 | 0.831 | 0.872 | 0.831 | 0.845 | 0.848 | 0.866 | 0.907 | 0.874 |
| | ImBD | 0.954 | 0.905 | 0.976 | 0.945 | 0.933 | 0.985 | 0.964 | 0.961 | 0.979 | 0.990 | 0.993 | 0.987 |
| | L2D | 0.999 | 0.955 | 0.995 | 0.983 | 0.982 | 0.997 | 0.980 | 0.986 | 0.984 | 0.999 | 0.997 | 0.993 |
| | *Abs. Gain (%)* | 4.5 | 5.0 | 1.9 | 3.8 | 4.9 | 1.2 | 1.6 | 2.6 | 0.4 | 0.9 | 0.4 | 0.6 |
| | *Rel. Gain (%)* | 97.8 | 53.0 | 81.2 | 69.6 | 73.3 | 79.4 | 44.8 | 65.4 | 21.4 | 87.1 | 62.4 | 46.4 |

Table 2 presents the AUC scores for all detectors across the 27 combinations of datasets, target models, and types of prompts. Our method achieves the best performance in nearly all cases, whereas ImBD (logits-based) or RAIDAR (rewrite-based) works as the second best. The relative gain over these best baselines is 70.11% on average, which again highlights (i) the advantage of rewrite-based methods over logits-based methods in settings with unseen prompts; and (ii) the effectiveness of learning an adaptive distance function over using a fixed distance in rewrite-based approaches.

## 4.3 EXPERIMENTS AGAINST ADVERSARIAL ATTACK

Following Bao et al. (2024), we further evaluate the robustness of our method against two types of adversarial attacks: (i) *Rephrasing*, where the LLM-written text is further paraphrased by a T5-based paraphraser before detection; (ii) *Decoherence*, where in each LLM-generated sentence containing more than 20 words, two adjacent words are randomly swapped. Both attacks are designed to reduce the coherence of LLM-generated text and have been shown to degrade the detection accuracy of existing detectors (Bao et al., 2024).

We conduct experiments on the same three datasets used in Section 4.2, resulting in a total of **six** settings. For comparison, we focus on ImBD and RAIDAR, as they achieve the second best performance on these datasets.

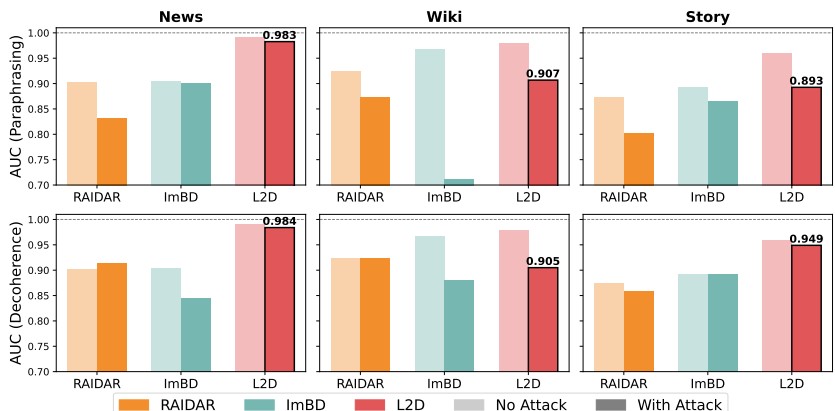

Figure 4: AUCs of ImBD, RAIDAR and our approach under paraphrasing (top panels) and decoherence (bottom panels). Each column represents a dataset. For each method, two bars are plotted: the lighter one indicates AUC without attack, and the darker one indicates AUC under attack. The best method under attack is highlighted with a bold bar edge, and its AUC value is displayed above the bar.

Figure 4 reports the AUC scores with and without adversarial attacks. While RAIDAR achieves comparable or superior AUCs on Story and Wiki in the absence of attacks, its AUC drops substantially under attacks, failing to maintain its lead. Similarly, ImBD's AUC declines considerably on Wiki under the rephrasing attack. In contrast, our method remains robust: its AUC either increases or remains unchanged on News, and only slightly decreases on other two datasets, achieving the best performance in each setting. This highlights the resilience of our approach to adversarial attacks and demonstrates its potential for reliable deployment in real-world scenarios.

### 4.4 ABLATION STUDY

We conduct an ablation study to compare against a version of our approach that uses the initial language model $p_\phi$ to construct the distance (FD, denoting a fixed distance). We consider the same settings to Section 4.2 and report the AUCs in Table B4. Our method consistently outperforms FD, with an average improvement of 97.1%. These results clearly demonstrate the advantage of learning the distance metric over fixing the distance.

## 5 DISCUSSION

This paper studies the detection of LLM-generated text. Our theoretical analysis offers geometric insights to demonstrate the effectiveness of rewrite-based approaches (Proposition 1) and their robustness to unseen prompts (Proposition 2). Methodologically, we go beyond existing rewrite-based methods by adaptively learning the distance function, which is theoretically grounded (Proposition 3) and delivers substantial empirical gains over both fixed-distance approaches (Section 4.4) and state-of-the-art detectors (Sections 4.1 and 4.2), while maintaining robustness against adversarial attacks (Section 4.3).

To conclude this paper, we remark that in our theoretical analysis, the assumptions were intentionally simplified (and thus stronger) to build geometric intuition behind these approaches. In Appendix A, we have offered a more complex version of our theories under less restrictive assumptions. Finally, although our method achieves state-of-the-art detection accuracy in most settings, its computational cost remains relatively high and comparable to existing rewrite-based algorithms (e.g., RAIDAR), due to the need to generate multiple rewrites (see Appendix B for detailed runtime results). This represents a potential limitation. We also note that asynchronous rewriting and distance computation using a vLLM backend can improve computational efficiency for practical deployment.

Several directions are worth investigating. It would be interesting to examine whether high-order reconstruction errors (Qi et al., 2026) can be used to enhance detection power. In addition, future work could explore whether the strong detection capability of L2D can be leveraged to identify LLM-generated segments in human–LLM co-authored texts (Li et al., 2026).

## ACKNOWLEDGEMENT

Hongyi Zhou and Ying Yang's research was partially supported by NSFC 12271286. The authors thank the anonymous referees and the area chair for their insightful and constructive comments, providing a significant improvement of the initial paper.

## ETHICS STATEMENT

The research presented in this paper adheres to the ICLR Code of Ethics (`https://iclr.cc/public/CodeOfEthics`) in all respects.

## REPRODUCIBILITY STATEMENT

We have made substantial efforts to ensure the reproducibility of this paper. The assumptions of our method are declared in Section 2, and the proofs of the theoretical results are provided in Appendix A. The implementation details of our approach (e.g., the choice of hyperparameters) are described in Appendix C. Additionally, the experimental setup and data generation procedures are explained in Section 4 and Appendix D. Together, these descriptions provide sufficient information for others to reproduce both our theoretical and empirical results.

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

# A    PROOFS AND ADDITIONAL THEORETICAL RESULTS

**Proof of Proposition 1:** We further assume $\mathcal{M}$ is a closed convex set so that the projection operator is well-defined. Then for any $x \in \mathcal{X}$ and $y \in \mathcal{M}$, we have

$$\langle x - \Pi_{\mathcal{M}}(x), y - \Pi_{\mathcal{M}}(x) \rangle \leq 0.$$

Taking $y = \mathcal{R}(x)$, it directly follows that

$$
\begin{aligned}
d^*(x, \mathcal{R}(x)) &= d^*(x, \mathcal{R}(x) - \Pi_{\mathcal{M}}(x) + \Pi_{\mathcal{M}}(x)) \\
&= d^*(x, \Pi_{\mathcal{M}_m}(x)) - 2\langle x - \Pi_{\mathcal{M}}(x), \mathcal{R}(x) - \Pi_{\mathcal{M}}(x) \rangle + |\mathcal{R}(x) - \Pi_{\mathcal{M}}(x)| \\
&\geq d^*(\Pi_{\mathcal{M}}(x), \mathcal{R}(x)) \quad \text{for all } x \in \mathcal{X}.
\end{aligned}
$$

Taking expectation on both sides with respect to $\boldsymbol{X} \sim p$, we obtain

$$\mathbb{E}_{\boldsymbol{X} \sim p}\{d^*(\boldsymbol{X}, \mathcal{R}(\boldsymbol{X}))\} \geq \mathbb{E}_{\boldsymbol{X} \sim p}\{d^*(\Pi_{\mathcal{M}}(\boldsymbol{X}), \mathcal{R}(\boldsymbol{X}))\} = \mathbb{E}_{\boldsymbol{X} \sim p}\{d^*(\Pi_{\mathcal{M}}(\boldsymbol{X}), \mathcal{R}(\Pi_{\mathcal{M}}(\boldsymbol{X})))\},$$

where the last equality follows from Assumption 3. Finally, Assumption 2 yields that

$$\mathbb{E}_{\boldsymbol{X} \sim p}\{d^*(\Pi_{\mathcal{M}}(\boldsymbol{X}), \mathcal{R}(\Pi_{\mathcal{M}}(\boldsymbol{X})))\} = \mathbb{E}_{\boldsymbol{X} \sim q}\{d^*(\boldsymbol{X}, \mathcal{R}(\boldsymbol{X}))\}.$$

Thus, the conclusion of Proposition 1 follows.

**Proof of Proposition 2:** According to the definition of projection operator $\Pi_{\mathcal{M}}$ and the fact that $\mathcal{R}(\boldsymbol{X})$ is supported on $\mathcal{M}$, it is obvious that

$$d^*(\boldsymbol{X}, \mathcal{R}(\boldsymbol{X})) \geq d^*(\boldsymbol{X}, \Pi_{\mathcal{M}}(\boldsymbol{X})). \tag{4}$$

Furthermore, the distribution of $q_{\text{prompt}}$ is also supported on $\mathcal{M}$. Therefore, combining equation equation 2, we obtain

$$
\begin{aligned}
\mathbb{E}_{\boldsymbol{X} \sim q_{prompt}}[d^*(\boldsymbol{X}, \mathcal{R}(\boldsymbol{X}))] &= \mathbb{E}_{\boldsymbol{X} \sim q_{prompt}}[d^*(\Pi_{\mathcal{M}}(\boldsymbol{X}), \mathcal{R}(\boldsymbol{X}))] \\
&= \mathbb{E}_{\boldsymbol{X} \sim q_{prompt}}[d^*(\Pi_{\mathcal{M}}(\boldsymbol{X}), \Pi_{\mathcal{M}}(\boldsymbol{X}) + e)] \\
&= \mathbb{E}_{\boldsymbol{X} \sim q_{prompt}}|e| \leq \epsilon. 
\end{aligned}
\tag{5}
$$

Combining inequality equation 4 and equation 5, the conclusion of Proposition 2 then follows.

**Additional Results**. The geometric assumptions in Section 2 were intentionally simplified to make our propositions interpretable. In fact, these assumptions could be relaxed to a more realistic setting. Specifically, we only assume

(i)  Human- and LLM-generated text lie on two nonlinear manifolds $\mathcal{H}$ and $\mathcal{M} \subseteq \mathcal{X}$, with their intrinsic dimensions $d_h > d_m$;

(ii)  Rewriting satisfies $\mathbb{E}[d^*(\mathcal{R}(x), x)] \leq \varepsilon_0$ for any $x \in \mathcal{M}$ and some small $0 < \varepsilon_0 < 1$, whereas $\sup_{x_1, x_2 \in \mathcal{M} \cup \mathcal{H}} d^*(x_1, x_2) = 1$;

(iii)  Human-written text distribution $p$ is absolutely continuous with respect to some $d_h$–dimensional volume measure $\mu$ on $\mathcal{H}$ with a bounded density.

Notice that (i) relaxes the linearity condition in Assumption 2 and does not assume that $\mathcal{M}$ is a projection or subspace of $\mathcal{H}$. Meanwhile, the assumption $d_h > d_m$ is well supported by empirical findings (Arora et al., 2023) which demonstrate that human text typically has intrinsic dimension of 8.5 - 10, whereas LLM-generated text has a dimension of only $6 - 8$ (Figure 1(c), Arora et al., 2023).

Furthermore, (ii) only requires that, for LLM-generated text, its reconstruction error is on average small relative to the maximum distance in the space. It does not require the error to be almost surely small as in the additive noise model, nor does it require equivalence in Assumption 3. In our empirical study, we find the ratio of this expected reconstruction error to the maximum distance is consistently very small across multiple datasets (see Table A1).

Under these realistic assumptions, we obtain the following proposition:

**Proposition.** Let $\kappa := d_h - d_m$. Under Assumptions (i)–(iii), for a human text $\boldsymbol{X}$ and an LLM-generated text $\boldsymbol{Y}$, the inequality

$$\mathbb{E}_{\widetilde{\boldsymbol{X}} \sim \mathcal{R}(\boldsymbol{X})}[d^*(\boldsymbol{X}, \widetilde{\boldsymbol{X}})] > \mathbb{E}_{\widetilde{\boldsymbol{Y}} \sim \mathcal{R}(\boldsymbol{Y})}[d^*(\boldsymbol{Y}, \widetilde{\boldsymbol{Y}})]$$

Table A1: Ratio of average reconstruction error of LLM-generated text to the maximum distance across different combinations of datasets and LLMs.

| Dataset | GPT-3-Turbo | GPT-4o | Gemini-1.5-Pro | Llama-3-70B |
|---|---|---|---|---|
| AcademicResearch | 0.065 | 0.074 | 0.074 | 0.059 |
| ArtCulture | 0.140 | 0.152 | 0.085 | 0.072 |
| Business | 0.114 | 0.073 | 0.048 | 0.078 |
| Code | 0.127 | 0.093 | 0.088 | 0.092 |
| EducationMaterial | 0.031 | 0.050 | 0.076 | 0.026 |
| Entertainment | 0.071 | 0.072 | 0.050 | 0.037 |
| Environmental | 0.057 | 0.060 | 0.034 | 0.052 |
| Finance | 0.084 | 0.139 | 0.042 | 0.053 |
| FoodCusine | 0.140 | 0.104 | 0.178 | 0.062 |
| GovernmentPublic | 0.112 | 0.097 | 0.047 | 0.054 |
| LegalDocument | 0.129 | 0.285 | 0.084 | 0.154 |
| LiteratureCreativeWriting | 0.060 | 0.070 | 0.037 | 0.048 |
| MedicalText | 0.163 | 0.169 | 0.069 | 0.107 |
| NewsArticle | 0.100 | 0.075 | 0.037 | 0.076 |
| OnlineContent | 0.138 | 0.207 | 0.105 | 0.049 |
| PersonalCommunication | 0.094 | 0.093 | 0.137 | 0.068 |
| ProductReview | 0.132 | 0.114 | 0.083 | 0.064 |
| Religious | 0.153 | 0.129 | 0.068 | 0.096 |
| Sports | 0.139 | 0.107 | 0.082 | 0.095 |
| TechnicalWriting | 0.082 | 0.083 | 0.033 | 0.043 |
| TravelTourism | 0.063 | 0.057 | 0.029 | 0.050 |

holds with probability at least $1 - O(\varepsilon_0^\kappa)$, where the expectations on both sides average out fluctuations in the rewriting process.

**Remark 1:** Given that empirical results suggest $\kappa$ is approximately 1.5 or 2 (Arora et al., 2023), the probability $1 - O(\varepsilon_0^\kappa)$ can be very close to 1 given that $\varepsilon_0$ is sufficiently small, which in turn proves that the reconstruction error for human-written text is, on average, larger than that for LLM-generated text.

**Remark 2:** The proof of the proposition relies on leveraging the assumption that $\mathcal{M}$ has a strictly lower intrinsic dimension than $\mathcal{H}$. Consequently, its $\varepsilon$−neighborhood overlaps with at most an $O(\varepsilon^\kappa)$ fraction of the human-text manifold. As a result, only a small proportion of human-written text lie within the $\varepsilon$−neighborhood of $\mathcal{M}$; most human text lie farther away, leading to the a larger reconstruction error.

**Proof:** Formally, for $\varepsilon > 0$, we denote the $\varepsilon_0$–tube (w.r.t. $d^\star$) around $\mathcal{M}$ as

$$\mathcal{N}_{\varepsilon_0}(\mathcal{M}) := \{x \in \mathcal{X} : \ d^*(x, \mathcal{M}) \leq \varepsilon_0\}.$$

Classical tube formulas imply

$$\mu\big(\mathcal{H} \cap \mathcal{N}_{\varepsilon_0}(\mathcal{M})\big) \ = \ O(\varepsilon_0^\kappa) \quad \text{as } \varepsilon_0 \downarrow 0.$$

Hence, under the bounded density assumption in (iii),

$$\mathbb{P}_{\boldsymbol{X} \sim p}\big\{d^*(\boldsymbol{X}, \mathcal{M}) < \varepsilon_0\big\} \ \leq \ C\,\mu\big(\mathcal{H} \cap \mathcal{N}_{\varepsilon_0}(\mathcal{M})\big) \ = \ O(\varepsilon_0^\kappa) \tag{6}$$

for some constant $C$. Therefore, with probability at least $1 - O(\varepsilon_0^\kappa)$,

$$\mathbb{E}_{\widetilde{\boldsymbol{X}} \sim \mathcal{R}(\boldsymbol{X})}[d^*(\boldsymbol{X}, \widetilde{\boldsymbol{X}})] - \mathbb{E}_{\widetilde{\boldsymbol{Y}} \sim \mathcal{R}(\boldsymbol{Y})}[d^*(\boldsymbol{Y}, \widetilde{\boldsymbol{Y}})] \geq d^*(\boldsymbol{X}, \mathcal{M}) - \varepsilon_0 > 0.$$

The proof is hence completed.

**Proof of Proposition 3:** Given that $d$ is bounded between $0$ and some positive constant $M$, we have $\mathbb{E}_{\boldsymbol{X} \sim p}[d(\boldsymbol{X}, \mathcal{R}(\boldsymbol{X})] \leq M$ and $\mathbb{E}_{\boldsymbol{X} \sim q_{\text{prompt}}}[d(\boldsymbol{X}, \mathcal{R}(\boldsymbol{X})] \geq 0$. Therefore, the reconstruction error is upper bounded by $M$. In what follows, we prove that by choosing $d = d_{opt}$, we can achieve this upper bound.

To prove this, we assume (i) – (iii) hold. As commented earlier, these assumptions are mild and are supported by empirical observations. Under these assumptions, letting the value of $\epsilon_0$ in equation 6 approach $0$, it follows that

$$\mathbb{P}_{\boldsymbol{X} \sim p}(\boldsymbol{X} \in \mathcal{M}) = 0.$$

Additionally, notice that the rewrite $\mathcal{R}(\boldsymbol{X})$ always lies in $\mathcal{M}$, it follows that

$$\mathbb{E}_{\boldsymbol{X}\sim p}[d_{opt}(\boldsymbol{X}, \mathcal{R}(\boldsymbol{X}))] = \mathbb{E}_{\boldsymbol{X}\sim p}[d_{opt}(\boldsymbol{X}, \mathcal{R}(\boldsymbol{X})\mathbb{I}(\boldsymbol{X} \in \mathcal{H}\backslash\mathcal{M})] = M.$$

Additionally, since $q$ is supported on $\mathcal{M}$, it follows that

$$\mathbb{E}_{\boldsymbol{X}\sim q_{\text{prompt}}}[d_{opt}(\boldsymbol{X}, \mathcal{R}(\boldsymbol{X}))] = 0.$$

Thus, under distance $d_{opt}$, the reconstruction error achieves the upper bound, which completes the proof.

# B ADDITIONAL IMPLEMENTATION DETAILS AND NUMERICAL EXPERIMENTS

We first provide an outline of our algorithm, which can be summarized into the following four steps:

1. Collect a dataset of human-authored text (denoted by $\mathcal{D}_h$) and prompt the target LLM (e.g., GPT-4o) to obtain an LLM-generated dataset (denoted by $\mathcal{D}_m$).

2. For each text $X \in \mathcal{D}_h \cup \mathcal{D}_m$, prompt an open-source lightweight LLM (specified below) to rewrite it $K$ times, and denoted the $K$ reconstructions by $\widetilde{\boldsymbol{X}}_1, \cdots, \widetilde{\boldsymbol{X}}_K$.

3. Learn a distance function $d_\phi$ that maximizes the difference in reconstruction errors between $\mathcal{D}_h$ and $\mathcal{D}_m$:

$$\max_{\phi}\ \mathbb{E}_{X\sim\mathcal{D}_h}\left[\frac{1}{K}\sum_{k=1}^{K}d_\phi(\boldsymbol{X}, \widetilde{\boldsymbol{X}}_k)\right] - \mathbb{E}_{X\sim\mathcal{D}_m}\left[\frac{1}{K}\sum_{k=1}^{K}d_\phi(\boldsymbol{X}, \widetilde{\boldsymbol{X}}_k)\right],$$

where $d_\phi(\boldsymbol{X}_1, X_2) = |\log p_\phi(\boldsymbol{X}_1)/|X_1| - \log p_\phi(\boldsymbol{X}_2)/|X_2||$ and $p_\phi$ is a language model whose architecture will be detailed below.

4. Given an input text $X$, obtain its reconstructions $\widetilde{\boldsymbol{X}}_1, \cdots, \widetilde{\boldsymbol{X}}_K$. If

$$\frac{1}{K}\sum_{k=1}^{K}d_\phi(\boldsymbol{X}, \widetilde{\boldsymbol{X}}_k),$$

exceeds a predefined threshold, classify $X$ as human-authored.

Table B1: AUC scores of various detectors for detecting text generated by GPT-4o.

| Dataset | Likelihood | LRR | IDE | BARTScore | FDGPT | Binoculars | RoBERTa | RADAR | ADGPT | RAIDAR | ImBD | L2D | AG (%) | RG (%) |
|---|---|---|---|---|---|---|---|---|---|---|---|---|---|---|
| AcademicResearch | 0.527 | 0.503 | 0.557 | 0.651 | 0.648 | 0.639 | 0.516 | 0.637 | 0.512 | 0.821 | 0.941 | 0.977 | 3.562 | 60.5 |
| ArtCulture | 0.500 | 0.518 | 0.504 | 0.638 | 0.590 | 0.605 | 0.570 | 0.560 | 0.605 | 0.660 | 0.762 | 0.871 | 10.918 | 45.8 |
| Business | 0.562 | 0.578 | 0.562 | 0.634 | 0.675 | 0.675 | 0.512 | 0.540 | 0.506 | 0.636 | 0.848 | 0.932 | 8.444 | 55.6 |
| Code | 0.563 | 0.641 | 0.551 | 0.646 | 0.681 | 0.679 | 0.589 | 0.554 | 0.502 | 0.605 | 0.806 | 0.932 | 12.580 | 64.8 |
| EducationMaterial | 0.643 | 0.806 | 0.611 | 0.825 | 0.800 | 0.754 | 0.724 | 0.746 | 0.583 | 0.952 | 0.997 | 0.996 | — | — |
| Entertainment | 0.694 | 0.659 | 0.595 | 0.846 | 0.826 | 0.818 | 0.668 | 0.793 | 0.525 | 0.855 | 0.982 | 0.993 | 1.039 | 58.6 |
| Environmental | 0.750 | 0.638 | 0.585 | 0.885 | 0.848 | 0.818 | 0.622 | 0.571 | 0.516 | 0.861 | 0.879 | 0.985 | 9.983 | 87.1 |
| Finance | 0.639 | 0.641 | 0.503 | 0.824 | 0.753 | 0.726 | 0.612 | 0.573 | 0.526 | 0.709 | 0.882 | 0.978 | 9.595 | 81.1 |
| FoodCusine | 0.625 | 0.542 | 0.535 | 0.783 | 0.719 | 0.699 | 0.558 | 0.507 | 0.512 | 0.703 | 0.915 | 0.969 | 5.476 | 64.1 |
| GovernmentPublic | 0.559 | 0.570 | 0.536 | 0.685 | 0.723 | 0.716 | 0.570 | 0.579 | 0.552 | 0.677 | 0.909 | 0.944 | 3.565 | 39.1 |
| LegalDocument | 0.523 | 0.527 | 0.622 | 0.700 | 0.690 | 0.689 | 0.528 | 0.547 | 0.555 | 0.630 | 0.971 | 0.939 | — | — |
| LiteratureCreativeWriting | 0.669 | 0.624 | 0.534 | 0.652 | 0.722 | 0.703 | 0.524 | 0.686 | 0.540 | 0.772 | 0.909 | 0.974 | 6.521 | 71.5 |
| MedicalText | 0.573 | 0.507 | 0.548 | 0.634 | 0.661 | 0.633 | 0.529 | 0.564 | 0.506 | 0.684 | 0.789 | 0.846 | 5.767 | 27.3 |
| NewsArticle | 0.512 | 0.578 | 0.529 | 0.600 | 0.605 | 0.603 | 0.515 | 0.784 | 0.517 | 0.785 | 0.902 | 0.986 | 8.394 | 85.4 |
| OnlineContent | 0.554 | 0.570 | 0.513 | 0.700 | 0.711 | 0.684 | 0.577 | 0.574 | 0.526 | 0.657 | 0.799 | 0.956 | 15.681 | 78.1 |
| PersonalCommunication | 0.539 | 0.520 | 0.000 | 0.571 | 0.623 | 0.616 | 0.511 | 0.518 | 0.515 | 0.598 | 0.670 | 0.873 | 20.381 | 61.7 |
| ProductReview | 0.682 | 0.670 | 0.512 | 0.804 | 0.740 | 0.731 | 0.583 | 0.544 | 0.538 | 0.691 | 0.893 | 0.977 | 8.398 | 78.4 |
| Religious | 0.666 | 0.593 | 0.566 | 0.892 | 0.521 | 0.509 | 0.585 | 0.763 | 0.557 | 0.725 | 0.969 | 0.990 | 2.025 | 66.2 |
| Sports | 0.564 | 0.511 | 0.515 | 0.565 | 0.641 | 0.644 | 0.507 | 0.556 | 0.506 | 0.681 | 0.828 | 0.903 | 7.534 | 43.7 |
| TechnicalWriting | 0.501 | 0.501 | 0.000 | 0.687 | 0.638 | 0.629 | 0.560 | 0.631 | 0.539 | 0.831 | 0.926 | 0.983 | 5.664 | 76.9 |
| TravelTourism | 0.501 | 0.501 | 0.539 | 0.687 | 0.638 | 0.629 | 0.560 | 0.631 | 0.540 | 0.795 | 0.939 | 0.985 | 4.521 | 74.6 |
| Average | 0.588 | 0.581 | 0.496 | 0.710 | 0.688 | 0.676 | 0.568 | 0.612 | 0.532 | 0.730 | 0.882 | 0.952 | 7.020 | 59.3 |
| Std | 0.072 | 0.075 | 0.164 | 0.099 | 0.077 | 0.071 | 0.054 | 0.088 | 0.026 | 0.093 | 0.080 | 0.043 | — | — |

Table B2: AUC scores of various detectors for detecting text generated by Llama-3-70B-Instruct.

| Dataset | Likelihood | LRR | IDE | BARTScore | FDGPT | Binoculars | RoBERTa | RADAR | ADGPT | RAIDAR | ImBD | L2D | AG (%) | RG (%) |
|---|---|---|---|---|---|---|---|---|---|---|---|---|---|---|
| AcademicResearch | 0.686 | 0.597 | 0.522 | 0.625 | 0.793 | 0.786 | 0.528 | 0.718 | 0.514 | 0.634 | 0.980 | 0.986 | 0.598 | 29.8 |
| ArtCulture | 0.643 | 0.635 | 0.643 | 0.640 | 0.829 | 0.835 | 0.538 | 0.586 | 0.626 | 0.630 | 0.902 | 0.945 | 4.302 | 43.7 |
| Business | 0.756 | 0.735 | 0.599 | 0.709 | 0.840 | 0.846 | 0.513 | 0.517 | 0.628 | 0.722 | 0.957 | 0.965 | 0.760 | 17.9 |
| Code | 0.554 | 0.631 | 0.574 | 0.620 | 0.765 | 0.761 | 0.556 | 0.621 | 0.561 | 0.723 | 0.886 | 0.951 | 6.421 | 56.5 |
| EducationMaterial | 0.841 | 0.912 | 0.583 | 0.914 | 0.936 | 0.919 | 0.565 | 0.903 | 0.538 | 0.627 | 0.999 | 0.999 | — | — |
| Entertainment | 0.933 | 0.815 | 0.587 | 0.940 | 0.979 | 0.978 | 0.802 | 0.862 | 0.590 | 0.629 | 0.999 | 1.000 | 0.092 | 100.0 |
| Environmental | 0.914 | 0.838 | 0.537 | 0.917 | 0.962 | 0.953 | 0.738 | 0.602 | 0.515 | 0.719 | 0.973 | 0.990 | 1.731 | 63.5 |
| Finance | 0.786 | 0.767 | 0.512 | 0.896 | 0.910 | 0.901 | 0.691 | 0.597 | 0.565 | 0.720 | 0.977 | 0.995 | 1.828 | 80.2 |
| FoodCusine | 0.800 | 0.698 | 0.569 | 0.827 | 0.854 | 0.843 | 0.556 | 0.542 | 0.551 | 0.629 | 0.978 | 0.999 | 2.111 | 94.0 |
| GovernmentPublic | 0.731 | 0.712 | 0.615 | 0.718 | 0.871 | 0.870 | 0.572 | 0.571 | 0.564 | 0.634 | 0.961 | 0.972 | 1.057 | 27.3 |
| LegalDocument | 0.503 | 0.662 | 0.589 | 0.763 | 0.884 | 0.876 | 0.517 | 0.696 | 0.607 | 0.720 | 0.990 | 0.972 | — | — |
| LiteratureCreativeWriting | 0.888 | 0.824 | 0.525 | 0.810 | 0.910 | 0.909 | 0.698 | 0.789 | 0.504 | 0.717 | 0.991 | 0.992 | 0.114 | 12.5 |
| MedicalText | 0.761 | 0.679 | 0.571 | 0.648 | 0.809 | 0.796 | 0.552 | 0.621 | 0.521 | 0.633 | 0.914 | 0.937 | 2.282 | 26.6 |
| NewsArticle | 0.688 | 0.583 | 0.563 | 0.652 | 0.839 | 0.826 | 0.643 | 0.857 | 0.631 | 0.629 | 0.973 | 0.994 | 2.118 | 78.9 |
| OnlineContent | 0.780 | 0.732 | 0.534 | 0.850 | 0.918 | 0.915 | 0.634 | 0.584 | 0.611 | 0.717 | 0.926 | 0.973 | 4.684 | 63.6 |
| PersonalCommunication | 0.691 | 0.625 | 0.590 | 0.607 | 0.770 | 0.761 | 0.535 | 0.522 | 0.596 | 0.718 | 0.838 | 0.950 | 11.199 | 69.3 |
| ProductReview | 0.873 | 0.769 | 0.545 | 0.870 | 0.872 | 0.863 | 0.583 | 0.546 | 0.544 | 0.632 | 0.983 | 0.996 | 1.366 | 78.7 |
| Religious | 0.599 | 0.505 | 0.506 | 0.927 | 0.740 | 0.724 | 0.559 | 0.814 | 0.617 | 0.729 | 0.995 | 0.943 | — | — |
| Sports | 0.699 | 0.600 | 0.667 | 0.506 | 0.789 | 0.788 | 0.522 | 0.573 | 0.558 | 0.720 | 0.952 | 0.939 | — | — |
| TechnicalWriting | 0.664 | 0.614 | 0.501 | 0.721 | 0.824 | 0.817 | 0.555 | 0.764 | 0.556 | 0.720 | 0.974 | 0.998 | 2.368 | 91.7 |
| TravelTourism | 0.664 | 0.614 | 0.501 | 0.721 | 0.824 | 0.817 | 0.555 | 0.764 | 0.510 | 0.634 | 0.982 | 0.996 | 1.346 | 75.4 |
| Average | 0.736 | 0.693 | 0.563 | 0.756 | 0.853 | 0.847 | 0.591 | 0.669 | 0.567 | 0.678 | 0.959 | 0.976 | 1.716 | 41.5 |
| Std | 0.113 | 0.099 | 0.045 | 0.125 | 0.064 | 0.065 | 0.078 | 0.121 | 0.041 | 0.045 | 0.041 | 0.022 | — | — |

In our experiments, the training and testing data differ in terms of models or data contexts. Specifically, in Tables 1 and B1, we train the distance function on text generated by GPT-4 and evaluate its performance to detect GPT-3.5-Turbo, and vice versa. In Table B3, we train the distance function on GPT-generated text but test it on text produced by Gemini. Thus, in all three tables, the training and testing models are either completely different or belong to the same family but correspond to different versions.

Moreover, all reported results therein are obtained via cross-fitting: we use one category of data (e.g., Story in Table 2) for testing and other categories (e.g., News and Wiki) for training. Consequently, the test data differ in content and domain from the training data.

Table B6 reports the average AUC and runtime of our method compared with RAIDAR, a state-of-the-art rewrite-based detector, in the setting of detecting text generated by GPT-3.5-Turbo (same to Table 1). As shown, our runtime is very close to that of RAIDAR – with only a slight increase – while achieving a substantial improvement in AUC. In addition, the reported runtime does not use a vLLM backend; incorporating vLLM could further reduce computational cost.

Table B3: AUC scores of various detectors for detecting text generated by Gemini 1.5 Pro.

| Dataset | Likelihood | LRR | IDE | BARTScore | FDGPT | Binoculars | RoBERTa | RADAR | ADGPT | RAIDAR | ImBD | L2D | AG (%) | RG (%) |
|---|---|---|---|---|---|---|---|---|---|---|---|---|---|---|
| AcademicResearch | 0.956 | 0.783 | 0.695 | 0.516 | 0.992 | 0.989 | 0.724 | 0.787 | 0.541 | 0.794 | 0.989 | 0.995 | 0.353 | 43.8 |
| ArtCulture | 0.807 | 0.774 | 0.890 | 0.586 | 0.982 | 0.975 | 0.862 | 0.506 | 0.664 | 0.577 | 0.913 | 0.955 | — | — |
| Business | 0.899 | 0.851 | 0.766 | 0.506 | 0.981 | 0.978 | 0.791 | 0.572 | 0.784 | 0.703 | 0.872 | 0.985 | 0.380 | 20.5 |
| Code | 0.567 | 0.670 | 0.683 | 0.618 | 0.829 | 0.805 | 0.842 | 0.585 | 0.579 | 0.567 | 0.820 | 0.979 | 13.736 | 86.9 |
| EducationMaterial | 0.998 | 0.989 | 0.607 | 0.871 | 1.000 | 1.000 | 0.889 | 0.911 | 0.859 | 0.968 | 1.000 | 1.000 | — | — |
| Entertainment | 0.995 | 0.916 | 0.689 | 0.860 | 1.000 | 1.000 | 0.625 | 0.911 | 0.863 | 0.927 | 1.000 | 1.000 | 0.020 | 80.0 |
| Environmental | 0.972 | 0.931 | 0.506 | 0.775 | 0.998 | 0.997 | 0.532 | 0.625 | 0.530 | 0.891 | 0.887 | 0.997 | — | — |
| Finance | 0.930 | 0.873 | 0.548 | 0.745 | 0.991 | 0.993 | 0.629 | 0.583 | 0.590 | 0.829 | 0.903 | 0.998 | 0.577 | 78.1 |
| FoodCusine | 0.794 | 0.608 | 0.566 | 0.552 | 0.901 | 0.895 | 0.573 | 0.594 | 0.572 | 0.791 | 0.992 | 0.986 | — | — |
| GovernmentPublic | 0.913 | 0.874 | 0.808 | 0.555 | 0.981 | 0.980 | 0.758 | 0.517 | 0.601 | 0.623 | 0.995 | 0.988 | — | — |
| LegalDocument | 0.578 | 0.847 | 0.644 | 0.520 | 0.998 | 0.998 | 0.952 | 0.917 | 0.615 | 0.683 | 0.983 | 1.000 | 0.162 | 100.0 |
| LiteratureCreativeWriting | 0.984 | 0.883 | 0.575 | 0.843 | 0.997 | 0.995 | 0.729 | 0.722 | 0.530 | 0.932 | 0.976 | 1.000 | 0.216 | 81.6 |
| MedicalText | 0.954 | 0.855 | 0.775 | 0.556 | 0.984 | 0.985 | 0.822 | 0.505 | 0.608 | 0.686 | 0.964 | 0.963 | — | — |
| NewsArticle | 0.911 | 0.705 | 0.612 | 0.617 | 0.987 | 0.991 | 0.538 | 0.926 | 0.810 | 0.827 | 0.998 | 0.999 | 0.018 | 10.7 |
| OnlineContent | 0.791 | 0.728 | 0.524 | 0.550 | 0.951 | 0.941 | 0.568 | 0.636 | 0.702 | 0.786 | 0.834 | 0.973 | 2.207 | 44.6 |
| PersonalCommunication | 0.813 | 0.678 | 0.582 | 0.559 | 0.870 | 0.872 | 0.682 | 0.632 | 0.598 | 0.782 | 0.591 | 0.950 | 7.778 | 60.7 |
| ProductReview | 0.888 | 0.730 | 0.541 | 0.589 | 0.959 | 0.958 | 0.509 | 0.663 | 0.629 | 0.765 | 0.990 | 0.995 | 0.503 | 49.4 |
| Religious | 0.558 | 0.551 | 0.613 | 0.850 | 0.873 | 0.856 | 0.854 | 0.805 | 0.737 | 0.854 | 0.961 | 0.996 | 3.477 | 89.3 |
| Sports | 0.811 | 0.667 | 0.795 | 0.799 | 0.934 | 0.929 | 0.772 | 0.560 | 0.597 | 0.694 | 0.808 | 0.965 | 3.110 | 47.3 |
| TechnicalWriting | 0.929 | 0.785 | 0.751 | 0.656 | 0.989 | 0.986 | 0.733 | 0.816 | 0.556 | 0.927 | 0.969 | 1.000 | 1.052 | 98.5 |
| TravelTourism | 0.929 | 0.785 | 0.751 | 0.656 | 0.989 | 0.986 | 0.733 | 0.816 | 0.532 | 0.851 | 0.994 | 0.998 | 0.371 | 63.2 |
| Average | 0.856 | 0.785 | 0.663 | 0.656 | 0.961 | 0.957 | 0.720 | 0.695 | 0.643 | 0.784 | 0.926 | 0.987 | 2.532 | 65.5 |
| Std | 0.134 | 0.110 | 0.106 | 0.125 | 0.049 | 0.054 | 0.126 | 0.143 | 0.105 | 0.114 | 0.097 | 0.016 | — | — |

Table B4: AUCs across 27 combinations of datasets, models, and prompt types, with the best method highlighted in cyan. The average absolute gain is 35.6%. The average relative gain over FD is 96.0%.

| Dataset | Method | Claude-3.5 | | | | GPT-4o | | | | Gemini | | | |
|---------|--------|---------|--------|--------|-------|---------|--------|--------|-------|---------|--------|--------|-------|
| | | rewrite | polish | expand | Avg. | rewrite | polish | expand | Avg. | rewrite | polish | expand | Avg. |
| News | FD | 0.541 | 0.539 | 0.576 | 0.552 | 0.525 | 0.515 | 0.579 | 0.540 | 0.576 | 0.613 | 0.645 | 0.611 |
| | L2D | 1.000 | 0.989 | 1.000 | 0.996 | 0.994 | 1.000 | 1.000 | 0.998 | 1.000 | 1.000 | 1.000 | 1.000 |
| Wiki | FD | 0.532 | 0.522 | 0.532 | 0.529 | 0.589 | 0.614 | 0.738 | 0.647 | 0.510 | 0.605 | 0.579 | 0.565 |
| | L2D | 0.955 | 0.942 | 0.953 | 0.950 | 0.963 | 0.987 | 0.993 | 0.981 | 0.983 | 0.982 | 0.988 | 0.984 |
| Story | FD | 0.612 | 0.647 | 0.728 | 0.662 | 0.683 | 0.821 | 0.892 | 0.799 | 0.641 | 0.800 | 0.856 | 0.766 |
| | L2D | 0.999 | 0.955 | 0.995 | 0.983 | 0.982 | 0.997 | 0.980 | 0.986 | 0.984 | 0.999 | 0.997 | 0.993 |

Table B5: Comparison between learning to rewriting (L2R) and our proposal. As L2R does not provides their implementations, we paste the results of Table 1 in Hao et al. (2025) into the Table. We can see that our proposal surpasses L2R in 20 datasets.

| Method | AcademicResearch | EducationMaterial | FoodCusine | MedicalText | ProductReview | TravelTourism | ArtCulture |
|--------|------------------|-------------------|------------|-------------|---------------|---------------|-----------|
| L2R | 0.8406 | 0.9644 | 0.9547 | 0.7857 | 0.9689 | 0.9475 | 0.8328 |
| L2D | 0.9885 | 0.9906 | 0.9907 | 0.9083 | 0.9948 | 0.9933 | 0.9204 |

| Method | Entertainment | GovernmentPublic | NewsArticle | Religious | LiteratureCreativeWriting | Environmental | LegalDocument |
|--------|---------------|------------------|-------------|-----------|---------------------------|---------------|---------------|
| L2R | 0.9494 | 0.8675 | 0.9242 | 0.9775 | 0.9294 | 0.9786 | 0.7803 |
| L2D | 0.9993 | 0.9620 | 0.9960 | 0.9656 | 0.9917 | 0.9902 | 0.9812 |

| Method | OnlineContent | Sports | Code | Finance | Business | PersonalCommunication | TechnicalWriting |
|--------|---------------|--------|------|---------|----------|-----------------------|------------------|
| L2R | 0.8881 | 0.8742 | 0.8383 | 0.9400 | 0.9156 | 0.8239 | 0.9369 |
| L2D | 0.9666 | 0.9308 | 0.9451 | 0.9912 | 0.9562 | 0.9334 | 0.9943 |

Table B6: Comparison of average AUC and runtime between RAIDAR and our method. Absolute AUC gain is computed as $(\text{AUC}_{\text{ours}} - \text{AUC}_{\text{RAIDAR}}) \times 100\%$ and relative AUC gain is computed as $(\text{AUC}_{\text{ours}} - \text{AUC}_{\text{RAIDAR}})/(1.0 - \text{AUC}_{\text{RAIDAR}}) \times 100\%$.

| Method | AUC | Runtime (s) | Gain (Abs. & Rel.) |
|--------|-----|-------------|--------------------|
| RAIDAR | 0.762 | 6.348 | – |
| L2D | 0.941 | 6.468 | 17.90% & 75.2% |

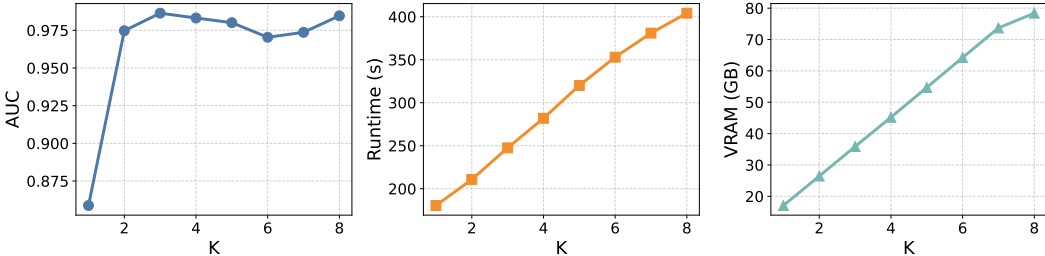

Figure B1: AUC, runtime for training, and memory usage during training when $K$ increases.

It is well known that varying the sampling temperature produces different outputs from LLMs, and adjusting temperature is a commonly used strategy in real-world LLM usage (Renze, 2024). In practice, when collecting text from an LLM, the specific temperature setting is typically unknown. It is therefore important to evaluate whether our method remains robust when training and test data are generated with different temperatures.

Following the same data generation process described in Section 4.3, we extend the setting to include six temperature values: $\{0.01, 0.2, 0.4, 0.6, 0.8, 1.0\}$. For evaluation, we partition the datasets into training and testing splits based on temperature. Specifically, one split uses $\{0.2, 0.6, 0.8\}$ for training and $\{0.01, 0.4, 1.0\}$ for testing, and the roles are reversed in the other split. This design mimics realistic scenarios where data collected at one set of temperatures are used to detect text generated at unseen temperatures.

As shown in Figure B2, our method achieves performance nearly identical to the case where training and test data share the same temperature. These results highlight the robustness of our approach under temperature variation.

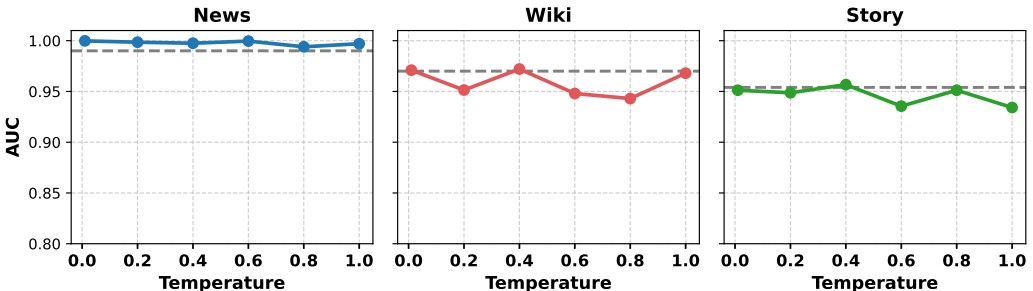

Figure B2: AUCs under varying temperatures. Each column corresponds to a dataset. Dashed lines indicate performance when training and test data are generated with the same temperature.

## C  IMPLEMENTATION

**Prompt for rewriting**. The prompt is set as: `You are a rewriting expert and you would rewrite the text without missing the original details. Return ONLY the rewritten version. Do not explain changes, do not give multiple options, and do not add commentary. Original text:`.

To generate rewritten texts, we employ an open-source model available on HuggingFace, i.e., `google/gemma-2-9b-it`. We recommend using an instruction fine-tuned variant, as it is more likely to produce faithful rewrite. In addition, the model should contain at least a billion parameters, since smaller models often fail to generate reliable rewrite. Choosing a open-source LLM does not require access to proprietary models like ChatGPT and Grok, making our approach being affordable and accessibility. We set the `max_new_tokens` as the 1.2 times of the number of tokens in $\boldsymbol{X}$, and the `min_new_tokens` as the 0.8 times of the number of tokens in $\boldsymbol{X}$.

**Rewrite times** $K$. The parameter $K$ plays a critical role in balancing computational cost and detection performance. Increasing $K$ improves the accuracy of estimating $\tau$, but at the expense of longer training time—since probabilities $p_\phi(\widetilde{\boldsymbol{X}}_1), \ldots, p_\phi(\widetilde{\boldsymbol{X}}_K)$ must all be computed—and higher GPU memory requirements during backpropagation. Figure B1 illustrates the trade-off: while larger $K$ generally improves performance, the gains diminish beyond small values, whereas the runtime and memory usage grow roughly linearly. Notably, as long as $K > 1$, the AUC remains strong. Motivated by this observation, we adopt a modest choice of $K = 4$ throughout all experiments, striking a balance between accuracy and efficiency.

**Fine-tuning setting**. In our specific fine-tuning, we set the distance function as $d_\phi(\boldsymbol{X}_1, \boldsymbol{X}_2) = |\log p_\phi(\boldsymbol{X}_1)/\texttt{len}(\boldsymbol{X}_1) - \log p_\phi(\boldsymbol{X}_2)/\texttt{len}(\boldsymbol{X}_2)|$ where $\texttt{len}(\boldsymbol{X}_k)$ is the number of tokens of $\boldsymbol{X}_k$ ($k = 1, 2$). This normalization accounts for text length, as a longer text are expected to correspond to smaller log-likelihood. Without loss of generality, we set $p_\phi$ as the model used for generating the rewritten text. We fine-tune the model, employ LoRA (Hu et al., 2022) implemented in the `peft` library, with rank parameter set to 8, lora_alpha set to 32, and lora_dropout set to 0.1, and the other parameters use the default settings.

## D  EXPERIMENTS: DETAILS

This section describes the experimental setup in detail. It is worth noting that throughout all experiments, we use AUC as the evaluation metric, and the relative gain over the strongest baseline is computed as: (Our AUC − StrongestBaseline's AUC)/(1.0 − StrongestBaseline's AUC).

### D.1 EXPERIMENTAL SETUP ON DIVERSE DATASETS

**Setup for learning-based methods**. For fairness, we follow a consistent training protocol across training-based detectors. Specifically, for each method, we train on 10 out of the 21 datasets and evaluate on the remaining ones. We then repeat the process by swapping the training and test splits, ensuring that no evaluation data leaks into training and guaranteeing a fair comparison. For *RoBERTa* and *RADAR*, since only pre-trained checkpoints are publicly available, we directly use the models released on HuggingFace[2][3]. This setup also enables a reasonable comparison with L2R, which uses 70% of each dataset for training and the remainder for testing. In contrast, our method trains on fewer datasets and the evaluation datasets are out of domains yet still achieves better performance, highlighting the effectiveness of the learning procedure.

**Setup for zero-shot methods**. For zero-shot detectors, we employ the same open-source LLMs as surrogate models to compute their statistical measures. These include *Likelihood*, *IDE*, and *LRR*. Notice that, the implementation of *IDE*[4] provide two method for estimating intrinsic dimension, one is based on persistence homology and another is based on maximum likelihood estimation (Levina & Bickel, 2004). Since the former requires a large amount of time on computing, we use maximum likelihood estimation in the experiments. For *Binoculars* and *FDGPT*, which require both a sampling model and a scoring model, we set $p_\phi$ as the scoring model and use its corresponding base model as the sampling model. For *BARTScore*, which also involves rewriting, we align its rewriting step with our own method while using the pre-trained BARTScore model from HuggingFace[5] to compute distances.

### D.2 EXPERIMENTAL SETUP ON DIFFERENT PROMPTS

**Data generation.** We generate machine-generated texts with three state-of-the-art LLMs: GPT-4o, Claude-3.5-Haiku, and Gemini-2.5-Flash. They specific version are: `gpt-4o-2024-08-06`, `claude-3-5-haiku-20241022`.

We next describe the specific system prompts and user prompts that are used for generating texts. First, for the *rewrite* task, the system prompt is:

> **System Prompt on Rewrite**
>
> You are a professional rewriting expert and you can help paraphrase this paragraph in English without missing the original details. Please keep the length of the rewritten text similar to the original text.

For the *polish* task, the system prompt is:

> **System Prompt on Polish**
>
> You are a professional polishing expert and you can help polish this paragraph.

For the *expand* task, the system prompt is:

> **System Prompt on Expand**
>
> You are a professional writing expert and you can help expand this paragraph.

For `Gemini-2.5-Flash` and `Claude-3.5-Haiku`, we additionally append the instruction in the system prompt:

---

[2]`https://huggingface.co/openai-community/roberta-large-openai-detector`
[3]`https://huggingface.co/TrustSafeAI/RADAR-Vicuna-7B`
[4]`https://github.com/ArGintum/GPTID`
[5]`https://huggingface.co/facebook/bart-large-cnn`

```
Return ONLY the rewritten/polished/expanded version.  Do not
explain changes, do not give multiple options, and do not add
commentary.
```

This ensures the output is strictly aligned with the assigned task.

The user prompt depends on the task. For rewriting, it takes the form: `Please rewrite:` `[a human text]`. For the expansion task, one of several predefined style prompts[6] is selected (e.g., "`Expand but not extend the paragraph in an oral style`" or "`Expand but not extend the paragraph in a literary style`"). For polishing, a prompt is similarly chosen from a predefined set[7] (e.g., "`Help me refine a paragraph with a lyrical touch.  Enhance the flow and imagery, making the words sing together in perfect harmony`").

Given these settings, each LLM generates texts from human-written texts randomly sampled from one of source datasets. In the generation process, we set the temperature parameter of LLM as 0.8. This process is repeated 100 times on one source dataset and one task, yielding a dataset of 100 machine-generated and 100 human-written texts. With three tasks, three LLMs, and three data sources, we obtain a total of 27 evaluation datasets.

**Setup of Baselines.** Baseline setups largely follow the procedure in Section D.1, with slight modifications to the training data. For instance, when evaluating performance on the *News* dataset, the *Wiki* and *Story* datasets are used for training. The process is repeated analogously when evaluating on the *Wiki* or *Story* datasets.

### D.3 Experimental Setup for Adversarial Attacks and Ablation

To evaluate the robustness of our approach against adversarial attacks, we adopt the attacks in Bao et al. (2024). In particular, for the rephrasing attack, we use the T5-based paraphraser available on HuggingFace[8] to paraphrase text generated by Claude-3.5 prior to detection.

In the ablation study, both FD and our method rely on the exact same rewritten texts to compute distance. This setup reflects the contribution of our adaptive distance learning procedure.

## E  Declaration: LLM usage

In preparing this paper, the LLM was used only for writing and editing, and it does not impact the core methodology.

---

[6]`https://github.com/Jiaqi-Chen-00/ImBD/blob/main/data/expand_prompt.json`

[7]`https://github.com/Jiaqi-Chen-00/ImBD/blob/main/data/polish_prompt.json`

[8]`https://huggingface.co/Vamsi/T5_Paraphrase_Paws`

