# OpenReview forum: "Learn-to-Distance: Distance Learning for Detecting LLM-Generated Text"
_ICLR.cc/2026/Conference — ICLR 2026 Poster_

### Official Review · Reviewer_J3u5 · 2025-10-27

**Soundness:** 2
**Presentation:** 2
**Contribution:** 2
**Rating:** 4
**Confidence:** 4

**Summary:**

This paper proposes a new rewrite-based detection algorithm for the detection problem of LLM generated text. The author first analyzes the principle and robustness of rewriting detection algorithms from a geometric perspective, and then proposes a new method for adaptive learning text rewriting distance.

**Strengths:**

1. Clear theoretical insight:
The geometric analysis (Propositions 1 & 2) provides an explanation for why rewrite-based approaches remain robust in unseen prompt scenarios.

2. Extensive experimentation:
The experiments are comprehensive — 27 settings across 3 datasets, 3 prompt types, and multiple target models (Claude-3.5, GPT-4o, Gemini-2.5) — showing consistent improvements.

3. Adversarial robustness:
The evaluation under paraphrasing and decoherence attacks demonstrates the proposed method’s strong robustness.

**Weaknesses:**

1. Questionable novelty of “Prompt-robust detection”:
The notion of prompt-robustness largely overlaps with prior work on generalization detection, where detectors are expected to perform under unknown prompts or semantic variations. Many existing zero-shot methods (e.g., DetectGPT, Fast-DetectGPT) are already prompt-agnostic, so it’s unclear whether the proposed notion is truly new.

2. Fairness of comparison (Figure 2):
The proposed method requires training, whereas Fast-DetectGPT is a zero-shot detector. Comparing them directly may be unfair since the proposed approach has access to additional supervision and compute.

3. Lack of implementation details:
The paper is vague about key training aspects, such as the architecture of the distance module, learning rate, optimization steps, dataset scale, or fine-tuning hyperparameters. This limits reproducibility.

4. High computational cost:
The method involves multiple LLM queries — one for rewriting and another for distance computation — which can be prohibitively expensive in large-scale or real-time scenarios.

5. Misleading improvement reporting:
The “relative gain” metric (e.g., “average improvements of 45.3%–62.5%”) can exaggerate the results. For example, when AUROC improves from 0.951 to 0.987, the absolute gain is minor (~0.036), yet the table reports a 72.9% “relative gain.” Reporting absolute AUC improvements would be more transparent and standard.

**Questions:**

1. What are the detailed training settings (loss function, optimizer, learning rate, batch size, training steps)?

2. Does Figure 2 involve an unfair comparison between a trained detector and zero-shot baselines?

---

> ### Author Response · Authors · 2025-11-20
>
> > Response D1: Novelty of "prompt-robust detection" (Q1)
>
> There may be some misunderstandings regarding our paper that could have led to your concern, and we appreciate the opportunity to clarify them in this rebuttal.
>
> Specifically, in our paper, *prompt-robust detection* refers to the robustness of a detection algorithm under settings where LLM-generated text is produced under various unseen prompts. While existing logits-based detectors such as Fast-DetectGPT can be applied to these settings, their robustness is limited. Indeed, the Fast-DetectGPT paper does include experiments with unseen prompts; however, those prompts are extremely short and drawn from a fixed set of types (see Appendix C.2 of their paper). In contrast, our study shows that these methods struggle with long and varied unseen prompts (see the first two panels of Figure 2). For example, in Figure 2, the unseen prompt involves a "polish" instruction combined with a paragraph of text, resulting in a substantially longer and more variable (due to the diversity of paragraph being polished) prompt.
>
> To the best of our knowledge, these settings have not been systematically addressed in the existing literature. For instance, existing rewrite-based methods typically employ a fixed distance, and as shown in the third panel of Figure 2, they fail to handle these settings effectively. This justifies our novelty. Additionally, two other reviewers (Reviewers oNqV and pN4E) explicitly appreciated our focus on such unseen-prompt settings, which reinforces the novelty of our work.
>
> > Response D2: Fairness in comparison (W2 & Q2)
>
> First, we have taken care to ensure fairness in all experimental comparisons, including those shown in Figure 2. For example, both zero-shot methods and our algorithm use the same base model for implementation. Additional procedures to guarantee fairness are detailed in our **Response C4** to Reviewer pN4E.
>
> Second, we agree with you that Fast-DetectGPT is training-free, which can result in certain disadvantages when compared to our algorithm. To address this, during this rebuttal, we implement AdaDetectGPT [1], a recently proposed variant of Fast-DetectGPT. This allows for a fair comparison, as both AdaDetectGPT and our algorithm require training. The results, reported in **Table B to Table D**, demonstrate that our method is more effective than AdaDetectGPT as well in distinguishing human text from LLM-generated text. If the paper is accepted, we would be happy to update the final panel of Figure 2 to use AdaDetectGPT rather than Fast-DetectGPT, should the referee prefer this.
>
> Third, during this rebuttal, we have conducted additional numerical studies under settings where the training and testing data differ in terms of models or data contexts, which reduces the inherent advantage of our algorithm over Fast-DetectGPT. The results, shown in **Table B to Table D** (see our response to Reviewer pN4E), demonstrate that our method continues to achieve considerably better performance than zero-shot methods.

---

> > ### Comment · Reviewer_J3u5 · 2025-11-26
> >
> > I appreciate the effort the authors have put into the rebuttal. However, after carefully reading your response, I feel that my core concerns remain largely unaddressed.
> >
> > First of all, for the first concern, I mean that the prompt robustness you proposed is similar to a lot of previous work in terms of concept and methods. In other words, it has been widely discussed.
> >
> > (1) Conceptual novelty
> >
> > In particular, it is not clear how your 'prompt robustness' is substantively different from semantic robustness/generalization. A prompt directly encodes the semantics of the desired output. For example, if I prompt a model to write a creative story or to generate a news article about a new event, the semantics of the generated text will naturally shift accordingly. In many realistic cases, the semantics expressed by the prompt (e.g., a new event) will not appear in the training data, so the model need to generalize to unseen semantic conditions. There is already a large body of work on semantic generalization/robustness in this setting.
> >
> > **In other words, prompt determines the semantics, so what you said about prompt robustness (in fact, I think it should be called generalization, because robustness is often aimed at attacks, but in fact, your original intention is not to attack, but to face the generalization performance of different prompts), which is essentially semantic generalization?**
> >
> > For instance, works such as DPIC[1], RAIDAR[2], and other zero-shot detection approaches explicitly discuss generalization beyond the training distribution. Note that the main contribution of Fast-DetectGPT is efficiency (hence “Fast”), but even that work clearly mentions that a key advantage of zero-shot methods lies in their generalization ability. Similarly, the DPIC paper explicitly argues that detector-based training suffers from poor generalization, and they propose prompt-based regeneration as a remedy to improve detector generalization. In addition, I want to emphasize that DPIC means **Decoupling *Prompt***, which already highlight the influence of prompts on generalization. This further reinforces my earlier point: the core idea in your work appears to have been extensively discussed in prior work.
> >
> > (2) Methodological novelty
> >
> > My second concern is about methodological novelty. The proposed approach—using multiple rewrites and comparing distances—appears very close in spirit to prior work such as RAIDAR, which also relies on rewriting-based techniques.
> >
> > Could the authors clearly articulate how their multi-rewriting strategy is substantially different from RAIDAR, beyond implementation details? If, in essence, the main change is just an improved or modified distance function, then I believe the current phrasing of the contribution may be overstated. In the paper, the “towards” framing suggests that prior work has not really explored this direction, and that you are the first to do so. Based on my understanding of RAIDAR, DPIC, and related work, this does not seem accurate.
> >
> > Last, in the AI text detection area, there are essentially two major lines of research:
> > - Generalization, which corresponds directly to what you describe as “prompt robustness.”
> > - Robustness to attacks, such as whether a detector can still identify generated text after paraphrasing or rewriting.
> >
> > Given that these two directions have already been well-established, I remain unclear on what the new conceptual insight is in your paper. Your framing suggests that prompt robustness is a new or previously under-explored idea, but this notion seems far from new.
> >
> > [1] DPIC: Decoupling Prompt and Intrinsic Characteristics for LLM Generated Text Detection
> >
> > [2] Raidar: geneRative AI Detection viA Rewriting
> >
> > ---
> >
> > In fact, I think the authors have done a relatively good job in the interpretability aspect of the work. If the main emphasis were placed on the distance metric itself, I actually believe there is room for novelty here. Prior works mostly rely on heuristic choices of distances and simply validate them empirically. If you could offer a deeper justification or theoretical motivation for your proposed distance function, that would indeed be a reasonable and meaningful contribution.
> >
> > However, the current version of the paper overstates its contributions, and this mismatch appears strongly in both the title and the narrative framing. In particular, the emphasis on “prompt robustness” is not a good choice, because this idea has already been extensively explored in prior work (e.g., DPIC, RADAR, and numerous zero-shot detection methods).

---

> ### Author Response · Authors · 2025-11-20
>
> > Response D3: Missing implementation details (W3 & Q1)
>
> To address this, we first provide an outline of our algorithm. We next detail our training implementation and answer your question. To further enhance reproducibility, we will make our complete workflow publicly available on GitHub/Hugging Face.
>
> First, our algorithm can be summarized into the following four steps:
>
> 1. Collect a dataset of human-authored text (denoted by $\mathcal{D}_h$) and prompt the target LLM (e.g., GPT-4o) to obtain an LLM-generated dataset (denoted by $\mathcal{D}_m$).
> 2. For each text $X\in \mathcal{D}_h\cup \mathcal{D}_m$, prompt an open-source lightweight LLM (specified below) to rewrite it $K$ times, and denoted the $K$ reconstructions by $\widetilde{X}_1,\cdots,\widetilde{X}_K$.
> 3. Learn a distance function $d_\phi$ that maximizes the difference in reconstruction errors between $\mathcal{D}_h$ and $\mathcal{D}_m$:
>
> $$\max\_\phi \mathbb{E}\_{X \sim \mathcal{D}\_h}\left[\frac{1}{K}\sum\_{k=1}^K d\_\phi(X, \widetilde{X}\_k) \right] - \mathbb{E}\_{X \sim \mathcal{D}\_m}\left[ \frac{1}{K}\sum\_{k=1}^K d_\phi(X, \widetilde{X}\_k) \right],$$
>
> where $d_\phi(X_1, X_2) = |  \log p_\phi(X_1)/|X_1| - \log p_\phi(X_2)/|X_2| |$ and $p_\phi$ is a language model whose architecture will be detailed below.
>
> 4. Given an input text $X$, obtain its reconstructions $\widetilde{X}_1,\cdots,\widetilde{X}_K$. If
> $$\frac{1}{K}\sum\_{k=1}^K d\_\phi(X, \widetilde{X}_k)$$
> exceeds a predefined threshold, classify $X$ as human-authored.
>
> Next, we detail our implementation. We use `google/gemma-2-9b-it` as the rewriting model, with $K=4$ in our experiments. For simplicity, the model $p_{\phi}$ in the distance function adopts the same architecture, `google/gemma-2-9b-it`. We train this distance function with a learning rate of $10^{-4}$, for 2 epochs, using a batch size of 1. The optimizer is AdamW combined with a CosineAnnealingLR scheduler. The size of the training dataset depends on the experimental setting: when more than 1000 training texts are available, we randomly select 1000 samples for training; otherwise, we use the maximum available number of training texts, typically fewer than 600 samples. Our fine-tuning hyperparameters were reported in Appendix C of our submission: We fine-tune the model using LoRA, implemented via the peft library, with rank $=$ 8, LoRA alpha $=$ 32, and LoRA dropout $=$ 0.1; all other parameters follow default settings.

---

> ### Author Response · Authors · 2025-11-20
>
> > Response D4: Computational cost (W4)
>
> You are right that our method queries the LLM twice -- once for rewriting and once for computing the distance -- but the actual computation time is faster than you might expect.
>
> Specifically, computing the distance is extremely fast, since it only requires to evaluate $\log p(\mathbf{X})$. This computation typically takes less than 0.2 seconds on a Hopper Architecture GPU. Notice that our method evaluates the log-likelihood $K+1$ times ($K$ rewritten texts, usually $K=4$, plus the original input), so the total time is less than 1 seconds.
>
> The main cost lies in the rewriting. However, this cost is also mild: generating a rewritten text for a moderate-length text usually takes less than 1 second. This is due to our use of a lightweight 9B model and vLLM backend for rewriting while achieving strong performance (for more details, refer to our **Response C3** to Reviewer pN4E). Additionally, we find that using only four rewrites is already sufficient to achieve high accuracy. This allows us to rewrite all samples in a single batch, keeping the cost low.
>
> To empirically demonstrate the feasibility of our algorithm, we report our quasi-online inference time in **Table F**. As shown, our algorithm completes within approximately 5 seconds for short texts ($<$1000 tokens), and even for long documents with more than 5,000 tokens (roughly 10-12 pages), the runtime remains under 15 seconds.
>
> Furthermore, we directly compare the average running times for our algorithm against FastDetectGPT (abbreviated as FDGPT) and RAIDAR in **Table E**. Among them, FastDetectGPT is logit-based and inherently the fastest but least accurate, whereas RAIDAR is a representative rewrite-based method requiring text generation. It is evident that all rewrite-based methods suffer from the computational bottleneck of rewriting. However, our method achieves a considerable improvement in AUROC with only a marginal increase in GPU time consumption (see **Table E**).
>
> Finally, we have further developed additional strategies for handling longer documents in our **Response C3** to Reviewer pN4E. These results collectively confirm the practicality of our algorithm for real-world deployment.
>
> **Table E**. AUC and Time Comparison. Here, the vLLM backend is incorporated to facilitate computation, resulting in a faster runtime for RAIDAR compared to its results in Table B5 of the Appendix. However, the runtime of our method in this table includes model-loading and other overheads. As shown in Table B5, once these overheads are removed, our runtime becomes very similar to that of RAIDAR.
> |Method|Avg. AUC|Avg. Time (s)/Sample|
> |----------|----------|-------------|
> |**FDGPT**|0.6047|0.0605|
> |**RAIDAR**|0.7618|5.8634|
> |**Ours**|0.9414|6.9680|
>
> **Table F**. Quasi-Online Inference (rewrite preparation + detection) time of our algorithm for input texts with varying number of tokens. "Count" indicates the number of texts within each token range. Note that reported times may overestimate real-world latency, as they include GPU model loading and offloading overheads which are avoidable in production environments.
>
> |Length Bin|Count|Avg Time (s)|Std Error (s)|Time/Token (ms)|
> |:-|:-|:-|:-|:-|
> |0-500|852|4.2770|0.0779|12.5773|
> |500-1000|1133|4.7186|0.0716|6.6503|
> |1000-1500|587|7.1101|0.1170|5.7157|
> |1500-2000|566|8.1558|0.0973|4.7308|
> |2000-2500|280|8.9435|0.1697|4.0074|
> |2500-3000|207|9.6250|0.1933|3.5114|
> |3000-3500|151|11.4709|0.2208|3.5366|
> |3500-4000|157|12.2581|0.1897|3.2961|
> |4000-4500|97|13.0363|0.2185|3.0740|
> |4500-5000|81|13.2825|0.2233|2.8177|
> |$\geq$5000|53|13.4623|0.3019|2.4916|
>
> > Response D5: "Relative gain" metric (W5).
>
> We first clarify that we did not invent this metric ourselves to exaggerate our results. It has been used in prior work, including the Fast-DetectGPT paper, to report the improvement of their detector. We adopted it here to maintain consistency with the existing literature.
>
> Second, if the paper is accepted, we will include both absolute AUC gains and relative improvements (with the detailed formula provided in the main text), to ensure that our improvements are presented clearly and transparently, without any exaggeration.
>
> **Reference**
>
> [1] Zhou, Hongyi, et al. ArXiv:2510.01268 (2025).

---

> ### Comment · Reviewer_J3u5 · 2025-11-26
>
> First, 1s is accpetable for calculating the distance. However, for one sample, **your method need about 7s.** ***This is clearly much slower than I expected.***
> Moreover, the use of vLLM cannot be cited as evidence of algorithmic speed, since vLLM is an engineering optimization that accelerates any model, not a property specific to your method.
>
> I would like to point out that speed is **a relative concept**. In practical deployment scenarios, your approach is significantly slower than most existing detection methods.
> Therefore, I do not think it is accurate to claim that computation cost is not a disadvantage. Relative to the majority of existing methods, the speed of your approach is clearly a disadvantage, and this should be acknowledged transparently rather than dismissed. A more honest framing would help position the method appropriately and avoid overstating its practical efficiency.
>
> Finally, since you explicitly acknowledge that rewriting-based approaches generally suffer from lower speed—and given that your method is in fact slower than RAIDAR, which is already known to be relatively slow—I believe this makes it even more important to openly acknowledge the computational cost as a real limitation.

---

> ### Comment · Reviewer_J3u5 · 2025-11-26
>
> The overhead of loading and unloading can also be completely avoided in your table. These overheads can be easily excluded when measuring runtime: you can simply load the model once and compute the inference time directly during the rewriting and detection steps. There is no need to time the entire execution of the script.
>
> Since you mention that the reported latency “does not reflect real-world deployment” due to including model loading/offloading time, then the paper should report the more realistic latency rather than the inflated one.
>
> ---
>
> These are my main concerns. I sincerely appreciate the effort the authors have put into the rebuttal. However, the current response does not fully address my questions and, in some aspects, leaves me even more uncertain. I would be glad to continue the discussion and work with the authors toward clarifying these issues and strengthening the overall quality of the paper.

---

> ### Author Response · Authors · 2025-12-03
>
> We sincerely thank the reviewer for the careful reading of our rebuttal and for the detailed, thoughtful comments. We have made every effort to address them, leading to a much improved paper; please refer to our response below and our revised manuscript (major changes are highlighted in red).
>
> > Response D6: Novelty and contributions
>
> We are grateful for the reviewer’s positive assessment of our theoretical interpretations. We fully agree that the two aspects the reviewer highlighted—the justification of rewrite-based detection methods and the proposed adaptively learned distance function—constitute the core contributions of our work. Following the reviewer’s suggestion, we have revised the manuscript to ensure that these contributions are presented clearly and without overstating novelty. Specifically, we have:
>
> 1. Revised the title by removing “towards” and “prompt-robust” to avoid implying that prompt robustness is previously unexplored, and adopted a title that emphasizes our adaptive learning of the distance function.
> 2. Added a new theorem (Proposition 3), along with several discussions following the proposition and Equation (3), to offer a deeper theoretical justification and motivation for our method.
> 3. Adjusted the framing by substantially reducing discussions centered on “prompt robustness,” keeping only minimal mentions and reframing them in terms of “generalization,” as suggested by the reviewer. We also revised the stated contributions to highlight the two key points above and addressed the mismatch noted by the reviewer.
>
> Additional details can be found in the revised manuscript. We note that the main text has not been substantially altered—our core ideas and methods remain unchanged. We have only added a single theorem and refined the framing to more clearly and accurately state our contributions and novelty.
>
> To conclude this point, we provide a concise summary of our theoretical results and how they motivate and justify our method (to aid the AC’s decision):
>
> - Proposition 1 (Page 4) provides geometric interpretations that reveal the rationale behind rewrite-based methods.
> - Proposition 2 (Page 5) demonstrates that these methods can generalize to unseen prompts.
> - Proposition 3 (Page 6) derives the form of the optimal distance function for implementing these methods; crucially, this function depends on the target language model to be detected.
> - Propositions 1 and 2 motivate considering rewrite-based detection, whereas Proposition 3 shows that methods relying on a fixed distance function cannot generalize well across different LLMs (see the discussion following Proposition 3).
> - We introduce a parameterized class of distance functions in Equation (3), directly motivated by Proposition 3, which can be viewed as a soft relaxation of the optimal distance derived there (see the discussion following Equation (3)).
> - We propose to learn the parameters of this distance function directly from data rather than relying on a pretrained model. This adaptive learning step is also theoretically motivated, and is empirically validated in Section 4.4 (see the discussion following Equation (3)).
>
> Altogether, these results justify our decision to adaptively learn the distance function.
>
> > Response D7: Computation
>
> We agree with the reviewer’s point. In response, we have:
>
> 1. Explicitly acknowledged that computational overhead is a potential limitation of our approach, despite its strong classification performance (Discussion Section, Page 10).
> 2. Mentioned the use of vLLM as a potential future direction for accelerating inference in practical deployment (Discussion Section, Page 10).
> 3. Reported computation times (excluding model loading and other overheads, and without using vLLM for acceleration) in Table B5 in Appendix B, showing that our method incurs only a marginal runtime increase over RAIDAR while achieving a much higher AUC.

---

### Official Review · Reviewer_pN4E · 2025-10-29

**Soundness:** 3
**Presentation:** 3
**Contribution:** 2
**Rating:** 4
**Confidence:** 3

**Summary:**

This paper proposes a prompt-robust detection framework for LLM-generated text based on a geometric understanding of rewrite-based methods. The authors show that human-written texts exhibit larger reconstruction errors than LLM-generated ones, and that this difference remains stable even under unseen prompts. Building on this insight, they introduce a method that learns an adaptive distance function between an input text and its rewritten version using a fine-tuned language model, rather than relying on fixed distances such as BLEU or BERTScore. Experiments across many datasets and several LLMs demonstrate improvement over baselines.

**Strengths:**

1. The paper is well organized and easy to follow
2. Important and timely topic
3. prompt robustness is very important for these trained llm-text classifiers.

**Weaknesses:**

1. My major concern lies in the geometric assumptions underpinning the theory, which are elegant but often unrealistic in practice. The framework assumes that LLM-generated text is a linear projection of human-written text onto an “LLM subspace” (Assumption 2) and that rewriting behaves equivalently on human and LLM-like inputs (Assumption 3). However, real-world text generation is highly nonlinear and context-dependent, and rewriting can amplify stylistic or semantic differences depending on prompts, temperature, and decoding randomness. The additive noise model  R(x) = \Pi_M(x) + e  also oversimplifies rewriting dynamics, as  e may not be small or confined to the same subspace. Consequently, the theoretical claims (Propositions 1–2) serve more as conceptual heuristics than as formal guarantees of robustness.

2. From a practical standpoint, the proposed approach faces several deployment challenges. First, it requires direct access to the target LLM for rewriting, which may not always be feasible in real-world . This dependency makes the method less suitable for general-purpose detection across unknown or evolving models. Second, the approach involves fine-tuning a distance model to adapt to each target LLM’s text distribution. This fine-tuning step also risks overfitting when the available training data are limited in domain diversity or prompt variety, potentially reducing generalization to unseen contexts. Finally, the paper offers little analysis on scalability and runtime efficiency, especially for long documents or large-scale batch detection scenarios where repeated rewriting and distance computation could become prohibitively expensive. These constraints limit the method’s immediate practicality despite its strong empirical performance.

3. In my understanding, your method requires fine-tuning the detector for each target LLM. Given this, is it fair to directly compare your approach with previous rewrite-based or other detection methods that operate in a zero-shot or non–fine-tuned setting? How do you ensure that the comparison across methods remains fair and consistent?

In general, I would consider this a borderline paper.

**Questions:**

1. The theoretical framework relies on strong geometric assumptions (e.g., LLM text as a projection of human text). Have you empirically verified or tested how well these assumptions hold in practice?

2. How well does the learned distance function transfer to new or unseen LLMs without re–fine-tuning?

---

> ### Author Response · Authors · 2025-11-20
>
> > Response C1: On geometric assumptions (W1 & Q1)
>
> Excellent point. We clarify that our intention was not to impose precise conditions for a mathematically rigorous analysis of rewrite-based detection methods. Rather, our assumptions were intentionally simplified (and thus stronger) to build geometric intuition behind these approaches. In general, there is a trade-off between mathematical precision and the interpretability of results, and in this work, we have prioritized the latter to inspire future research along these lines, both theoretical and empirical.
>
> That said, we acknowledge that the currently imposed assumptions may be overly idealized. In response to your comment, we have prepared a more mathematically precise — albeit more complex — version of our theories, which we will incorporate into the revised manuscript (e.g., in the Discussion and Appendix).
>
> In this more refined version, we **_no longer require_**:
>
> - LLM-generated text forms a linear subspace representing a projection of human-written text (Assumption 2).
> - Rewriting behaves equivalently on human- and LLM-generated text (Assumption 3).
> - The additive noise model assumption.
>
> To the contrary, we only assume:
>
> (i) Human- and LLM-generated text lie on two nonlinear manifolds $\mathcal{H}$ and $\mathcal{M} \subseteq \mathcal{X}$, with their intrinsic dimensions $d_h > d_m$;
> (ii) Rewriting satisfies $\mathbb{E}[d^*(\mathcal{R}(x), x)] \le \varepsilon_0$ for any $x \in \mathcal{M}$ and some small $0 < \varepsilon_0 < 1$, whereas
>
> $\sup_{x_1, x_2 \in \mathcal{M} \cup \mathcal{H}} d^*(x_1, x_2) = 1$;
>
> (iii) Human-written text distribution $p$ is absolutely continuous with respect to some $d_h$–dimensional volume measure $\mu$ on $\mathcal H$ with a bounded density.
>
> Notice that (i) relaxes the linearity condition in Assumption 2 and does not assume that $\mathcal{M}$ is a projection or subspace of $\mathcal{H}$. Meanwhile, the assumption $d_h > d_m$ is well supported by empirical findings (e.g., [1]), which demonstrate that human text typically has intrinsic dimension of **8.5–10**, whereas LLM-generated text has a dimension of only **6–8**.
>
> Furthermore, (ii) only requires that, for LLM-generated text, its reconstruction error is on average small relative to the maximum distance in the space. It does not require the error to be almost surely small as in the additive noise model, nor does it require equivalence in Assumption 3. In our empirical study, we find the ratio of this expected reconstruction error to the maximum distance is consistently very small across multiple datasets (see **Table G** in our response to Reviewer oNqV).
>
> Under these realistic assumptions, we obtain the following proposition:
>
> **Proposition.**
> Let $\kappa := d_h - d_m$. Under Assumptions (i)–(iii), for a human text $X$ and an LLM-generated text $Y$, the inequality
>
> $$\mathbb{E}\_{\tilde{X}\sim \mathcal{R}(X)}[d^\star(X,\tilde{X})] > \mathbb{E}\_{\tilde{Y}\sim \mathcal{R}(Y)}[d^\star(Y,\tilde{Y})]$$
>
> holds with probability at least $1 - O(\varepsilon_0^\kappa)$, where the expectations on both sides average out fluctuations in the rewriting process.
>
> **Remark 1.**
> Given that empirical results suggest $\kappa$ is approximately 1.5 or 2 [1], the probability $1-O(\varepsilon_0^\kappa)$  can be close to 1 when $\varepsilon_0$ is sufficiently small, which proves that the reconstruction error for human-written text is, on average, larger than that for LLM-generated text.
>
> **Remark 2.**
> The proof relies on leveraging the assumption that $\mathcal{M}$ has a strictly lower intrinsic dimension than $\mathcal{H}$. Consequently, its $\varepsilon$-neighborhood overlaps with at most an $O(\varepsilon^\kappa)$ fraction of the human-text manifold. As a result, only a small proportion of human-written text lies within the $\varepsilon$-neighborhood of $\mathcal{M}$; most human text lies farther away, leading to a larger reconstruction error.
>
> **Proof.**
> For $\varepsilon_0 > 0$, denote the $\varepsilon_0$–tube (w.r.t. $d^*$) around $\mathcal{M}$ as:
>
> $\mathcal{N}_{\varepsilon_0}(\mathcal{M}) :=  \{ x \in \mathcal{X} : d^*(x, \mathcal{M}) \leq \varepsilon_0 \}$.
>
> Classical tube formulas imply:
>
> $\mu(\mathcal{H} \cap \mathcal{N}_{\varepsilon_0}(\mathcal{M})) = O(\varepsilon_0^\kappa) \quad \text{as } \varepsilon_0 \downarrow 0.$
>
> Hence, under the bounded density assumption in (iii):
>
> $\mathbb{P}\_{X \sim p} \{d^*(X, \mathcal{M}) < \varepsilon_0\} \leq C \mu(\mathcal{H} \cap \mathcal{N}_{\varepsilon_0}(\mathcal{M}))= O(\varepsilon_0^\kappa)$
>
> for some constant $C$. Therefore, with probability at least $1 - O(\varepsilon_0^\kappa)$,
>
> $$\mathbb{E}\_{\tilde{X}\sim \mathcal{R}(X)}[d^\star(X,\tilde{X})] -  \mathbb{E}\_{\tilde{Y}\sim \mathcal{R}(Y)}[d^\star(Y,\tilde{Y})] \geq d^\star(X,\mathcal{M})-\varepsilon_0>0$$
>
> The proof is hence completed.

---

> ### Author Response · Authors · 2025-11-20
>
> > Response C2: Transfer to new/unseen LLMs without re-fine-tuning (W2 & Q2)
>
> Another excellent point. The second weakness you commented — along with your second question — raises four concerns:
> (i) direct access to the target LLM for rewriting;
> (ii) detecting unseen LLMs;
> (iii) overfitting during training that reduces generalization to unseen contexts;
> (iv) scalability and runtime efficiency.
> We address the first three concerns in this response and provide a discussion of (iv) in the following bullet point.
>
> For (i), we clarify that our algorithm does *not* need to use the target LLM for rewriting. In our numerical implementation (Section 4), we employ a lightweight model, `google/gemma-2-9b-it`, to detect text generated by GPT, Gemini, and Claude.
>
> For (ii), we argue that our learned distance function continues to achieve strong performance on unseen LLMs without re–fine-tuning. To support this claim, we designed three additional experimental settings during the rebuttal, with either entirely different training and testing models or models from the same series but different versions. Specifically:
>
> - In the first two settings, we train the distance function on text generated by GPT-4 and evaluate its ability to detect GPT-3.5-Turbo, and vice versa.
> - In the third setting, we train the distance function using GPT-generated text but test it for detecting Gemini.
>
> We report the numerical results in **Table B to Table D**. It can be seen that our algorithm continues to outperform the strongest baseline, achieving AUCs over 0.9 in most cases. This demonstrates the strong generalization capabilities of the learned distance function across different language models.
>
> For (iii), our results in (ii) already demonstrate that training does not overly reduce generalization to unseen models. Similarly, the empirical study in our main paper demonstrates that training does not reduce generalization to unseen contexts. Specifically, all reported results therein are obtained via **cross-fitting**: we use one category of data (e.g., Story) for testing and other categories (e.g., News and Wiki) for training. Consequently, the test data differ in content and domain from the training data. As shown in our numerical study, the proposed algorithm achieves the best performance in most cases, confirming that overfitting does not overly reduce generalization to unseen contexts.
>
> To conclude, we remark that this generalization arises from the design of our objective, which learns a distance function that
> (a) assigns small reconstruction errors to LLM-generated text, and
> (b) assigns large reconstruction errors to human-written text.
>
> This is clearly illustrated in Figure 2 of the main text: human-written text achieves reconstruction errors in the range of **40–60**, whereas LLM-generated text typically has reconstruction errors **smaller than 5**. Although switching to another LLM may slightly increase the reconstruction error for machine-generated text, the distance for human-written text remains substantially larger. Similarly, when switching to different contexts, the gap narrows but remains significant. Consequently, the learned distance function continues to deliver strong performance across models and datasets.

---

> ### Author Response · Authors · 2025-11-20
>
> > Response C3: Scalability, long documents & large-scale batch detection (W2)
>
> Regarding scalability, as mentioned in our previous response, we employ a lightweight LLM for rewriting, which facilitates computation while achieving high classification accuracy. Further details on runtime can be found in our **Response D4** to Reviewer J3u5. In the rest of this response, we address your comments concerning long documents and large-scale batch detection.
>
> Handling long documents is perhaps less computationally intensive than you might expect. Our key observation is that both the theoretical results in [2] and empirical findings in [3] suggest that using more tokens for detection generally improves performance. This trend is also evident in our experiments (see **Table A**): when applying our algorithm to the News, Wiki, and Story datasets, its AUC consistently increases with the number of words used for detection. However, the improvement becomes marginal once the word count exceeds 160.
>
> These results motivate the following procedure for handling long documents:
> - We first count the number of words in the passage.
> - If it exceeds a threshold (e.g., **500 words**), we classify the text as long and avoid using the full document for evaluation.
> - Instead, we use only the first 500 words and apply our algorithm accordingly.
>
> Alternatively,
> - We divide the entire text into **500-word segments**,
> - Apply our algorithm to each segment,
> - And classify the document as human-authored *only if all* segments are classified as human-authored.
>
> These procedures are substantially more computationally efficient while maintaining accuracy.
>
> Regarding large-scale batch detection, our current implementation already incorporates efficient batch processing (detailed in our **Response D4** to Reviewer J3u5). Specifically, we utilize asynchronous rewriting and distance computation with **vLLM backend** for practical deployment. Finally, scalability can be further enhanced through balanced routing within clusters.
>
> > Response C4: Fairness in comparison with zero-shot detectors (W3)
>
> We address this point from three perspectives:
>
> **First**, we have taken care to ensure fairness in all experimental comparisons. Specifically:
> (i) Both the zero-shot methods and our approach use the same base model, `google/gemma-2-9b-it`, as the rewrite and/or scoring model.
> (ii) When comparing against other learning-based methods such as RADIAR and ImBD, we also adopt this model to maintain consistency.
> (iii) For each input text, we use the same set of rewritten texts across all rewrite-based algorithms to ensure a fair comparison.
> (iv) For algorithms like ImBD that involve fine-tuning, we use the same optimization hyperparameters (e.g., number of epochs, learning rate) across all cases to ensure fairness in training.
>
> **Second**, we recognize that zero-shot methods such as Fast-DetectGPT are training-free; however, this advantage often comes at the cost of reduced accuracy compared to ML-based methods. To address this, during this rebuttal, we additionally implement **AdaDetectGPT** [2], a recently proposed variant of Fast-DetectGPT. This allows for a fair comparison, as both AdaDetectGPT and our algorithm require training. Our empirical results — reported in **Table B to Table D** (AdaDetectGPT is abbreviated as **ADGPT**) — show that our method consistently outperforms AdaDetectGPT.
>
> **Third**, our newly obtained results in **Table B to Table D** and numerical findings in the main paper further alleviate this concern. In these experiments, the training and testing data differ in terms of models or data contexts, which reduces the inherent advantage of ML-based approaches over zero-shot methods. Even under these shifts, our method continues to achieve the best performance in most cases and remains considerably better than zero-shot methods.
>
> **References**
>
> [1] S. Arora, S. Geer, R. Li, and F. Zelaya. *Intrinsic Dimension Estimation for Robust Detection of AI-Generated Texts.* arXiv preprint arXiv:2306.04723, 2023.
>
> [2] Zhou, H., Zhu, J., Su, P., Ye, K., Yang, Y., Gavioli-Akilagun, S. A., & Shi, C. (2025). *AdaDetectGPT: Adaptive detection of LLM-generated text with statistical guarantees.* arXiv preprint arXiv:2510.01268.
>
> [3] Bao, G., Zhao, Y., Teng, Z., Yang, L., & Zhang, Y. (2023). *Fast-DetectGPT: Efficient zero-shot detection of machine-generated text via conditional probability curvature.* arXiv preprint arXiv:2310.05130.
>
> **Table A.** AUCs of our detector on three datasets with varying numbers of input tokens.
>
> |Dataset|5|10|20|40|80|160|320|
> |-|-|-|-|-|-|-|-|
> |News|0.631|0.716|0.837|0.969|0.999|1.000|1.000|
> |Wiki|0.591|0.764|0.896|0.965|0.996|1.000|0.992|
> |Story|0.614|0.764|0.869|0.963|0.977|0.992|0.999|

---

> ### Author Response · Authors · 2025-11-20
>
> **Table B.** AUROC scores of various detectors for detecting text generated by GPT-3.5 Turbo. We use `google/gemma-2-9b-it` as the rewriting and scoring model for implementing both rewrite- and logits-based methods. The best result is marked in **bold** and the second-best in *italic*.
>
> |Dataset|Likelihood|LRR|IDE|BARTScore|FDGPT|Binoculars|RoBERTa|RADAR|ADGPT|RAIDAR|ImBD|Ours|
> |-|-|-|-|-|-|-|-|-|-|-|-|-|
> |AcademicResearch|0.582|0.557|0.571|0.561|0.542|0.532|0.510|0.718|0.544|0.812|*0.919*|**0.948**|
> |ArtCulture|0.529|0.539|0.508|0.620|0.556|0.580|0.605|0.618|0.549|0.618|*0.732*|**0.835**|
> |Business|0.532|0.563|0.574|0.639|0.657|0.656|0.564|0.587|0.518|0.704|*0.861*|**0.914**|
> |Code|0.677|0.530|0.601|0.551|0.556|0.568|0.525|0.702|0.575|0.539|*0.771*|**0.906**|
> |EducationMaterial|0.561|0.813|0.705|0.808|0.785|0.707|0.708|0.847|0.557|0.961|**0.996**|*0.973*|
> |Entertainment|0.601|0.645|0.725|0.866|0.805|0.745|0.750|0.887|0.510|0.875|**0.983**|*0.982*|
> |Environmental|0.672|0.636|0.608|0.854|0.830|0.770|0.680|0.647|0.569|0.850|*0.932*|**0.984**|
> |Finance|0.546|0.608|0.618|0.819|0.730|0.699|0.678|0.647|0.507|0.750|*0.956*|**0.987**|
> |FoodCusine|0.569|0.534|0.524|0.739|0.639|0.625|0.562|0.526|0.569|0.735|*0.869*|**0.969**|
> |GovernmentPublic|0.530|0.551|0.572|0.680|0.697|0.692|0.612|0.639|0.531|0.748|*0.903*|**0.923**|
> |LegalDocument|0.740|0.509|0.807|0.637|0.741|0.701|0.596|0.819|0.503|0.595|*0.991*|**0.994**|
> |LiteratureCreativeWriting|0.541|0.520|0.705|0.645|0.634|0.550|0.637|0.866|0.653|0.784|*0.993*|**0.996**|
> |MedicalText|0.553|0.564|0.538|0.591|0.620|0.600|0.519|0.629|0.556|0.654|*0.754*|**0.828**|
> |NewsArticle|0.655|0.674|0.656|0.555|0.513|0.506|0.626|0.861|0.616|0.785|*0.893*|**0.968**|
> |OnlineContent|0.539|0.525|0.512|0.711|0.654|0.632|0.596|0.604|0.541|0.743|*0.844*|**0.950**|
> |PersonalCommunication|0.555|0.521|0.515|0.602|0.541|0.547|0.526|0.581|0.555|0.653|*0.755*|**0.922**|
> |ProductReview|0.625|0.628|0.553|0.803|0.688|0.675|0.611|0.591|0.529|0.728|*0.880*|**0.971**|
> |Religious|0.741|0.642|0.662|0.884|0.534|0.543|0.579|0.869|0.648|0.812|**0.970**|*0.957*|
> |Sports|0.511|0.531|0.510|0.522|0.584|0.592|0.561|0.606|0.527|0.664|*0.821*|**0.910**|
> |TechnicalWriting|0.594|0.559|0.569|0.594|0.555|0.537|0.516|0.739|0.519|0.818|*0.944*|**0.994**|
> |TravelTourism|0.590|0.538|0.571|0.600|0.550|0.525|0.531|0.741|0.503|0.824|*0.917*|**0.989**|
> |Average|0.593|0.580|0.600|0.680|0.639|0.618|0.595|0.701|0.551|0.745|*0.890*|**0.948**|
> |Std|0.066|0.071|0.080|0.113|0.095|0.078|0.066|0.112|0.042|0.099|0.082|0.047|
>
> **Table C.** AUROC scores of various detectors for detecting text generated by GPT-4o. The setup is the same the above table.
>
> |Dataset|Likelihood|LRR|IDE|BARTScore|FDGPT|Binoculars|RoBERTa|RADAR|ADGPT|RAIDAR|ImBD|Ours|
> |-|-|-|-|-|-|-|-|-|-|-|-|-|
> |AcademicResearch|0.527|0.503|0.557|0.651|0.648|0.639|0.516|0.637|0.512|0.821|*0.941*|**0.977**|
> |ArtCulture|0.500|0.518|0.504|0.638|0.590|0.605|0.570|0.560|0.605|0.660|*0.762*|**0.871**|
> |Business|0.562|0.578|0.562|0.634|0.675|0.675|0.512|0.540|0.506|0.636|*0.848*|**0.932**|
> |Code|0.563|0.641|0.551|0.646|0.681|0.679|0.589|0.554|0.502|0.605|*0.806*|**0.932**|
> |EducationMaterial|0.643|0.806|0.611|0.825|0.800|0.754|0.724|0.746|0.583|0.952|**0.997**|*0.996*|
> |Entertainment|0.694|0.659|0.595|0.846|0.826|0.818|0.668|0.793|0.525|0.855|*0.982*|**0.993**|
> |Environmental|0.750|0.638|0.585|*0.885*|0.848|0.818|0.622|0.571|0.516|0.861|0.879|**0.985**|
> |Finance|0.639|0.641|0.503|0.824|0.753|0.726|0.612|0.573|0.526|0.709|*0.882*|**0.978**|
> |FoodCusine|0.625|0.542|0.535|0.783|0.719|0.699|0.558|0.507|0.512|0.703|*0.915*|**0.969**|
> |GovernmentPublic|0.559|0.570|0.536|0.685|0.723|0.716|0.570|0.579|0.552|0.677|*0.909*|**0.944**|
> |LegalDocument|0.523|0.527|0.622|0.700|0.690|0.689|0.528|0.547|0.555|0.630|**0.971**|*0.939*|
> |LiteratureCreativeWriting|0.669|0.624|0.534|0.652|0.722|0.703|0.524|0.686|0.540|0.772|*0.909*|**0.974**|
> |MedicalText|0.573|0.507|0.548|0.634|0.661|0.633|0.529|0.564|0.506|0.684|*0.789*|**0.846**|
> |NewsArticle|0.512|0.578|0.529|0.600|0.605|0.603|0.515|0.784|0.517|0.785|*0.902*|**0.986**|
> |OnlineContent|0.554|0.570|0.513|0.700|0.711|0.684|0.577|0.574|0.526|0.657|*0.799*|**0.956**|
> |PersonalCommunication|0.539|0.520|0.000|0.571|0.623|0.616|0.511|0.518|0.515|0.598|*0.670*|**0.873**|
> |ProductReview|0.682|0.670|0.512|0.804|0.740|0.731|0.583|0.544|0.538|0.691|*0.893*|**0.977**|
> |Religious|0.666|0.593|0.566|0.892|0.521|0.509|0.585|0.763|0.557|0.725|*0.969*|**0.990**|
> |Sports|0.564|0.511|0.515|0.565|0.641|0.644|0.507|0.556|0.506|0.681|*0.828*|**0.903**|
> |TechnicalWriting|0.501|0.501|0.000|0.687|0.638|0.629|0.560|0.631|0.539|0.831|*0.926*|**0.983**|
> |TravelTourism|0.501|0.501|0.539|0.687|0.638|0.629|0.560|0.631|0.540|0.795|*0.939*|**0.985**|
> |Average|0.588|0.581|0.496|0.710|0.688|0.676|0.568|0.612|0.532|0.730|*0.882*|**0.952**|
> |Std|0.072|0.075|0.164|0.099|0.077|0.071|0.054|0.088|0.026|0.093|0.080|0.043|

---

> ### Author Response · Authors · 2025-11-20
>
> **Table D**. AUROC scores of various detectors for detecting text generated by Gemini 1.5 Pro. We use `google/gemma-2-9b-it` as the rewriting and scoring model for implementing both rewrite- and logits-based methods. The highest scores are highlighted in **bold**, the second best in *italic*.
>
> |Dataset|Likelihood|LRR|IDE|BARTScore|FDGPT|Binoculars|RoBERTa|RADAR|ADGPT|RAIDAR|ImBD|Ours|
> |:-|:-|:-|:-|:-|:-|:-|:-|:-|:-|:-|:-|:-|
> |AcademicResearch|0.956|0.783|0.695|0.516|*0.992*|0.989|0.724|0.787|0.541|0.794|0.989|**0.995**|
> |ArtCulture|0.807|0.774|0.890|0.586|**0.982**|*0.975*|0.862|0.506|0.664|0.577|0.913|0.955|
> |Business|0.899|0.851|0.766|0.506|*0.981*|0.978|0.791|0.572|0.784|0.703|0.872|**0.985**|
> |Code|0.567|0.670|0.683|0.618|0.829|0.805|*0.842*|0.585|0.579|0.567|0.820|**0.979**|
> |EducationMaterial|0.998|0.989|0.607|0.871|**1.000**|**1.000**|0.889|0.911|0.859|0.968|**1.000**|**1.000**|
> |Entertainment|0.995|0.916|0.689|0.860|**1.000**|**1.000**|0.625|0.911|0.863|0.927|**1.000**|**1.000**|
> |Environmental|0.972|0.931|0.506|0.775|**0.998**|*0.997*|0.532|0.625|0.530|0.891|0.887|*0.997*|
> |Finance|0.930|0.873|0.548|0.745|0.991|*0.993*|0.629|0.583|0.590|0.829|0.903|**0.998**|
> |FoodCusine|0.794|0.608|0.566|0.552|0.901|0.895|0.573|0.594|0.572|0.791|**0.992**|*0.986*|
> |GovernmentPublic|0.913|0.874|0.808|0.555|0.981|0.980|0.758|0.517|0.601|0.623|**0.995**|*0.988*|
> |LegalDocument|0.578|0.847|0.644|0.520|0.998|*0.998*|0.952|0.917|0.615|0.683|0.983|**1.000**|
> |LiteratureCreativeWriting|0.984|0.883|0.575|0.843|*0.997*|0.995|0.729|0.722|0.530|0.932|0.976|**1.000**|
> |MedicalText|0.954|0.855|0.775|0.556|*0.984*|**0.985**|0.822|0.505|0.608|0.686|0.964|0.963|
> |NewsArticle|0.911|0.705|0.612|0.617|0.987|0.991|0.538|0.926|0.810|0.827|*0.998*|**0.999**|
> |OnlineContent|0.791|0.728|0.524|0.550|*0.951*|0.941|0.568|0.636|0.702|0.786|0.834|**0.973**|
> |PersonalCommunication|0.813|0.678|0.582|0.559|0.870|*0.872*|0.682|0.632|0.598|0.782|0.591|**0.950**|
> |ProductReview|0.888|0.730|0.541|0.589|0.959|0.958|0.509|0.663|0.629|0.765|*0.990*|**0.995**|
> |Religious|0.558|0.551|0.613|0.850|0.873|0.856|0.854|0.805|0.737|0.854|*0.961*|**0.996**|
> |Sports|0.811|0.667|0.795|0.799|*0.934*|0.929|0.772|0.560|0.597|0.694|0.808|**0.965**|
> |TechnicalWriting|0.929|0.785|0.751|0.656|*0.989*|0.986|0.733|0.816|0.556|0.927|0.969|**1.000**|
> |TravelTourism|0.929|0.785|0.751|0.656|0.989|0.986|0.733|0.816|0.532|0.851|*0.994*|**0.998**|
> |Average|0.856|0.785|0.663|0.656|*0.961*|0.957|0.720|0.695|0.643|0.784|0.926|**0.987**|
> |Std|0.134|0.110|0.106|0.125|0.049|0.054|0.126|0.143|0.105|0.114|0.097|0.016|

---

> > ### Comment · Reviewer_pN4E · 2025-11-23
> >
> > Thanks for your detailed rebuttal. It addresses part of my concerns. However, I still feel that the newly introduced geometric assumptions differ substantially from the original submission, and this appears to introduce significant modifications to the initial paper. Regarding the second point, the authors have addressed most of my concerns. For the fairness aspect, I hope the authors will incorporate these clarifications into the revised version.
> >
> > I intend to keep my original score, however, I will not object if the paper is ultimately accepted.

---

> ### Author Response · Authors · 2025-11-26
>
> > Response C5
>
> Dear reviewer,
>
> We are glad to hear that your second point has been mostly addressed. As for fairness, following your suggestion, we have incorporated the relevant clarifications into the revised paper (see Sections 4 and 5, where the major changes are highlighted in red).
>
> Based on your comments, your remaining concern applies only to the difference between our newly introduced assumptions and original ones. As we mentioned in our response, our goal is to keep the theory simple and intuitive to encourage future research. For this reason, we have kept both versions in the paper. The new version is now discussed in the Discussion section and presented in the Appendix for mathematical rigor. This keeps the changes to the main paper minimal.
>
> We hope this addresses your concern, and we would be happy to make further adjustments if needed.
>
> Best,

---

### Official Review · Reviewer_oNqV · 2025-10-30

**Soundness:** 4
**Presentation:** 3
**Contribution:** 4
**Rating:** 8
**Confidence:** 2

**Summary:**

This paper addresses the prompt-robust detection of LLM-generated text, a realistic and challenging setting where the prompts used to produce the text are unobserved. The authors propose a geometric interpretation of rewrite-based detection methods, proving why reconstruction errors differ between human and machine text (Proposition 1) and remain robust to prompt-induced distribution shifts (Proposition 2). Building on this, they introduce a machine-learning-based rewrite detector that learns a distance function via fine-tuning a language model, rather than relying on fixed metrics like BERTScore or Levenshtein distance. Extensive experiments across 24 datasets, 7 LLMs, and 3 prompt types show relative AUC improvements of 45–62% over the strongest baselines, with strong resistance to adversarial paraphrasing and decoherence attacks.

**Strengths:**

- Introduces a clear geometric interpretation of rewrite-based detectors, offering theoretical insight that previous empirical works lacked.
- The learned-distance formulation bridges theory and implementation elegantly, and the optimization is compatible with LoRA-style fine-tuning for scalability.
- Evaluated on > 100 settings (24 datasets, 7 LLMs, 3 prompt types, 2 attacks) with consistent superiority over 11 baselines.
- Addresses the realistic "unseen-prompt" condition that undermines most prior detectors, and demonstrates resilience under paraphrasing/decoherence.

**Weaknesses:**

- The Hilbert-space and projection assumptions (Assumptions 1–3) are strong, but empirical verification of these geometric hypotheses is limited.
- The authors mention small declines on certain datasets but do not analyze why the learned distance struggles there.
- Because the detector is trained using LLM-generated corpora, it may implicitly learn stylistic or semantic regularities specific to those generation distributions. Discussing how well the method generalizes beyond the seven tested generators (e.g., to unseen LLM families) would strengthen the paper.
- Fine-tuning a surrogate model to learn the distance is non-trivial; runtime and parameter-efficiency trade-offs are not quantified.

**Questions:**

Although this is a well-motivated paper that advances in prompt-robust LLM detection, I still have a few questions that I would truly appreciate if the authors could kindly address during the rebuttal:
- The experiments convincingly demonstrate robustness to paraphrasing and decoherence attacks. Could the authors kindly comment on how the method might behave under stronger semantic-preserving perturbations (e.g., back-translation, style transfer), or whether they foresee any limitations under such conditions?
- While the average AUC gains are impressive, the tables do not include variance or confidence intervals. Would it be possible for the authors to share whether these results were averaged over multiple random seeds, and how stable the improvements are across runs?
- Finally, since the approach requires multiple rewritings and fine-tuning, could the authors give a sense of the computational cost? For example, how much GPU time is typically needed?

---

> ### Author Response · Authors · 2025-11-20
>
> > Response B1: Assumption \& empirical validation (W1)
>
> As mentioned in our **Response C1** to Reviewer pN4E, the assumptions in the main text are intentionally simplified (and thus stronger) to build geometric intuition and to inspire future research on rewrite-based methods, both theoretical and empirical.
>
> To address your concern, we have prepared a revised version of our theoretical results with much less restrictive assumptions, at the cost of losing some straightforward interpretation. If our paper is accepted, we plan to include both versions.
>
> The detailed assumptions and theoretical developments are provided in our **Response C1** to Reviewer pN4E. Here, we briefly summarize the changes to the assumptions: (i) We now allow LLM- and human-authored text to reside on nonlinear manifolds rather than linear subspaces as in the original assumptions; (ii) We relax the projectional, equivalence, and additive assumptions; (iii) We impose two new assumptions:
> (a) the intrinsic dimensionality of the human-authored text manifold is larger than that of the LLM-generated text manifold; (b) the reconstruction error of LLM-generated text is small relative to the maximum distance in the space.
>
> We next validate our new assumptions (a) and (b):
>
> (a) is well supported by empirical findings [3] which demonstrate that human text typically has intrinsic dimension of 8.5 - 10, whereas LLM-generated text has a dimension of only 6 – 8 (Figure 1(c) of [3]).
>
> To verify (b), we have conducted an empirical study during the rebuttal by computing the ratio of the average reconstruction error of LLM-generated text to the maximum distance across various combinations of datasets and LLMs, and report them in **Table G**. It can be seen that this ratio is typically below 0.15 and, in most cases, can be as low as 0.05.
>
> **Table G**. Ratio of average reconstruction error of LLM-generated text to the maximum distance across different combinations of datasets and LLMs.
>
> |Dataset|GPT-3-Turbo|GPT-4o|Gemini-1.5-Pro|Llama-3-70B|
> |:-|:-|:-|:-|:-|
> |AcademicResearch|0.065|0.074|0.074|0.059|
> |ArtCulture|0.140|0.152|0.085|0.072|
> |Business|0.114|0.073|0.048|0.078|
> |Code|0.127|0.093|0.088|0.092|
> |EducationMaterial|0.031|0.050|0.076|0.026|
> |Entertainment|0.071|0.072|0.050|0.037|
> |Environmental|0.057|0.060|0.034|0.052|
> |Finance|0.084|0.139|0.042|0.053|
> |FoodCusine|0.140|0.104|0.178|0.062|
> |GovernmentPublic|0.112|0.097|0.047|0.054|
> |LegalDocument|0.129|0.285|0.084|0.154|
> |LiteratureCreativeWriting|0.060|0.070|0.037|0.048|
> |MedicalText|0.163|0.169|0.069|0.107|
> |NewsArticle|0.100|0.075|0.037|0.076|
> |OnlineContent|0.138|0.207|0.105|0.049|
> |PersonalCommunication|0.094|0.093|0.137|0.068|
> |ProductReview|0.132|0.114|0.083|0.064|
> |Religious|0.153|0.129|0.068|0.096|
> |Sports|0.139|0.107|0.082|0.095|
> |TechnicalWriting|0.082|0.083|0.033|0.043|
> |TravelTourism|0.063|0.057|0.029|0.050|
>
> > Response B2: Small declines on certain datasets (W2)
>
> This is an excellent comment. Investigating cases where our performance declines is valuable for understanding why our method struggles and identifying approaches for improvement. During the rebuttal, we have investigated these cases, analyzed the underlying reasons, and addressed them -- resulting in a further improved method.
>
> In total, there are two cases where these declines occur: (a) Section 4.2 of the main paper, where the baseline detectors outperform ours in a few instances (see Table 2); and (b) Section 4.3, where our performance decreases under adversarial attacks (see Figure 4).
>
> *Case (a)*. We investigated the rewritten text and the reconstruction errors of LLM-generated and human-authored texts in these cases. We found that the issue arises from the quality of the rewritten text. Specifically, in these cases, the rewritten text occasionally includes additional comments that are irrelevant to the original content being detected, such as:
>
> "** Changes Made: ** * Improved sentence structure and flow for better readability. * Used more varied vocabulary to enhance the text's richness. * Added transition words and phrases to create a smoother narrative. * Emphasized key details and arguments for clarity. The rewritten text retains all the original information while presenting it in a more engaging and polished manner. ".
>
> These unnecessary comments in the rewritten texts inflate the reconstruction error for LLM-generated text, causing our method to underperform relative to non-rewriting-based approaches such as ImBD. To address this, we refined the rewriting prompt to eliminate such comments and re-ran the experiments, which resulted in a nearly uniform improvement across almost all cases (see **Table H**). Regarding the remaining two cases where our method slightly underperforms ImBD, we believe this is due to the nature of the “expand” task in the prompt -- this task is relatively simple, and ImBD already achieves strong performance on it.

---

> ### Author Response · Authors · 2025-11-20
>
> We also investigated other potential factors, such as model architecture. For instance, in the main paper, we used `google/gemma-2-9b-it` as the rewriting and scoring model. During the rebuttal, we replaced it with `mistralai/Mistral-7B-Instruct-v0.3` and `Qwen/Qwen2.5-7B-Instruct`, and reported the results in **Table I** and **Table J**. While some cases still show our algorithm underperforming ImBD, these cases are not exactly the same as those in Table 1 and Table B1, indicating that performance declines also depend on the choice of model architecture.
>
> Finally, we note that the rewriting configuration (prompt template, number of rewrites, decoding temperature, etc.) in our current implementation was fixed uniformly across all datasets. This “one-size-fits-all” setting is not necessarily optimal, and we expect that employing domain-specific prompts and hyperparameters could further improve the performance of our detector. We plan to explore this direction in future work.
>
> *Case (b)*. Under adversarial attacks, LLM-generated text is paraphrased or intentionally perturbed to resemble human-written text (e.g., by introducing errors or noise). Such perturbations increase the reconstruction error for LLM-generated text, reducing the gap between LLM and human texts and making them harder to distinguish. This explains why our method may exhibit a slight decline on adversarially attacked datasets. This phenomenon has been widely reported in the literature, where paraphrasing and decoherence strategies are shown to effectively evade AI-text detectors [1]. Nevertheless, as discussed in Section 4.3, our method demonstrates stronger resilience to adversarial attacks compared to existing approaches.
>
> > Response B3: Unseen LLMs (W3).
>
> This is another excellent comment. During the rebuttal, we conducted additional experiments to assess the performance of our algorithm on LLMs that were not seen during training. In these experiments, the training and testing models are either entirely different or from the same series but different versions. The results, presented in **Table B to Table D** (see our response to Reviewer pN4E), show that our method continues to achieve superior performance for detecting text generated by previously unseen LLMs. Further details can be found in our **Response C2** to Reviewer pN4E.
>
> > Response B4: Computational cost for training and inference (W4 & Q3)
>
> We report the quasi-online inference time of our algorithm in **Table F** (see our response to Reviewer J3u5), which includes the time for rewriting and computing the distance. It can be seen that our inference procedure has a modest computational cost: it can process text with 500 tokens within 4 seconds, which is acceptable for most practical applications. Furthermore, for long documents (e.g., those with more than 3,000 tokens), the computation time is typically under 15 seconds. Refer to our **Response D4** to Reviewer J3u5 for further details.
>
> The training time is typically within one hour. This moderate training time is due to two factors: (i) we adopt LoRA for fine-tuning, which requires updating only about 1% of the parameters; and (ii) we find that using approximately 1,000 samples is sufficient to achieve optimal performance, allowing us to keep the sample size small and complete training quickly. As you noted, there is a trade-off between runtime and parameter efficiency. For example, fine-tuning the entire model without LoRA and increasing the training sample size could further improve detection performance, but at the cost of higher computational time.
>
> > Response B5: Robustness to stronger semantic-preserving perturbations (Q1)
>
> Excellent point. It is worthwhile to investigate stronger semantic-preserving perturbations, such as back-translation and style transfer, for assessing the robustness of the proposed algorithm. However, rewrite-based methods are generally less robust to these attacks. For example, consider back-translation (see [2]): the attack on human-written text first translates it from e.g., English to Chinese and then back to English. The resulting text is already generated by language models, which decreases the reconstruction error for attacked human-authored text and makes it extremely difficult to distinguish from LLM-generated content. We acknowledge this as an important direction for future research.

---

> ### Author Response · Authors · 2025-11-20
>
> > Response B6: Variance or confidence intervals (Q2)
>
> Good point. We clarify that our experiments were conducted on real datasets, and for each dataset, we performed only a single run. However, since we evaluated our method across multiple datasets (e.g., 21 datasets for Tables 1 and B1–B3 in our submission), the variance or confidence intervals are computed across these datasets, rather than across repeated runs on a single dataset as is common in traditional simulation studies.
>
> Indeed, Tables 1 and B1–B3 in our submission already report the standard deviation of the average AUCs across 21 datasets from different domains. During the rebuttal, we also added the standard deviation for the AUCs in Table 2 of the main text (see **Table K**). These results confirm that our proposed method achieves *statistically significant* improvements over the baselines.
>
> **Reference**
>
> [1] Hu, Xiaomeng, et al. NeurIPS (2023).
>
> [2] Wu, Junchao, et al. NeurIPS (2024).
>
> [3] Tulchinskii, Eduard, et al. NeurIPS (2023).
>
> **Table H**. AUROC across datasets, models, and tasks; best method highlighted in **bold**, second best in *italic*, and relative gain of Ours over the best baseline shown.
>
> |Dataset|Method|Claude(rewrite)|Claude(polish)|Claude(expand)|Claude(Avg.)|GPT(rewrite)|GPT(polish)|GPT(expand)|GPT(Avg.)|Gemini(rewrite)|Gemini(polish)|Gemini(expand)|Gemini(Avg.)|
> |:-|:-|:-|:-|:-|:-|:-|:-|:-|:-|:-|:-|:-|:-|
> ||Likelihood|0.598|0.604|0.645|0.616|0.572|0.587|0.539|0.566|0.594|0.579|0.732|0.635|
> ||LRR|0.594|0.626|0.636|0.619|0.633|0.620|0.559|0.604|0.656|0.601|0.717|0.658|
> ||Binoculars|0.555|0.634|0.709|0.633|0.535|0.567|0.631|0.578|0.507|0.632|0.589|0.576|
> ||IDE|0.606|0.686|0.726|0.673|0.577|0.736|0.696|0.670|0.608|0.672|0.716|0.665|
> ||FDGPT|0.524|0.610|0.686|0.607|0.508|0.561|0.641|0.570|0.507|0.617|0.586|0.570|
> ||BARTScore|0.728|0.583|0.563|0.625|0.653|0.526|0.549|0.576|0.567|0.606|0.671|0.615|
> |**News**|RoBERTa|0.544|0.524|0.546|0.538|0.509|0.532|0.568|0.536|0.501|0.566|0.567|0.545|
> ||RADAR|0.744|0.805|0.912|0.821|0.774|0.966|0.994|0.911|0.807|0.858|0.920|0.862|
> ||RAIDAR|0.912|0.885|0.926|0.908|0.867|0.891|0.873|0.877|0.864|0.882|0.949|0.898|
> ||ImBD|*0.941*|*0.928*|*0.990*|*0.953*|*0.966*|*0.999*|*0.999*|*0.988*|*0.937*|*0.977*|*0.990*|*0.968*|
> ||Ours|**1.000**|**0.995**|**1.000**|**0.998**|**1.000**|**1.000**|**1.000**|**1.000**|**1.000**|**1.000**|**1.000**|**1.000**|
> ||Ours(old)|1.000|0.990|1.000|0.997|1.000|1.000|1.000|1.000|1.000|1.000|1.000|1.000|
> |||||||||||||||
> ||Likelihood|0.519|0.532|0.562|0.538|0.546|0.553|0.649|0.583|0.505|0.512|0.533|0.517|
> ||LRR|0.532|0.508|0.540|0.527|0.541|0.612|0.695|0.616|0.522|0.508|0.536|0.522|
> ||Binoculars|0.608|0.667|0.762|0.679|0.619|0.717|0.862|0.733|0.571|0.768|0.793|0.711|
> ||IDE|0.565|0.621|0.613|0.600|0.584|0.712|0.682|0.659|0.573|0.642|0.699|0.638|
> ||FDGPT|0.587|0.646|0.739|0.658|0.597|0.712|0.867|0.725|0.557|0.748|0.791|0.699|
> ||BARTScore|0.760|0.634|0.520|0.638|0.785|0.592|0.529|0.635|0.605|0.590|0.615|0.603|
> |**Wiki**|RoBERTa|0.635|0.659|0.759|0.684|0.565|0.590|0.522|0.559|0.638|0.740|0.782|0.720|
> ||RADAR|0.533|0.507|0.620|0.553|0.541|0.814|0.933|0.763|0.550|0.564|0.680|0.598|
> ||RAIDAR|*0.926*|*0.936*|0.919|*0.927*|0.854|0.853|0.877|0.861|0.859|0.918|0.953|0.910|
> ||ImBD|0.913|0.931|*0.968*|*0.937*|*0.904*|*0.979*|**0.995**|*0.959*|*0.940*|*0.966*|**0.987**|*0.965*|
> ||Ours|**0.979**|**0.977**|**0.973**|**0.976**|**0.983**|**0.993**|*0.990*|**0.989**|**0.981**|**0.982**|*0.986*|**0.983**|
> ||Ours(old)|0.976|0.970|0.969|0.972|0.954|0.983|0.988|0.975|0.961|0.963|0.970|0.965|
> |||||||||||||||
> ||Likelihood|0.502|0.532|0.587|0.541|0.623|0.740|0.814|0.725|0.512|0.656|0.702|0.623|
> ||LRR|0.556|0.540|0.596|0.564|0.570|0.728|0.739|0.679|0.504|0.563|0.632|0.566|
> ||Binoculars|0.595|0.663|0.755|0.671|0.674|0.739|0.806|0.740|0.624|0.832|0.927|0.794|
> ||IDE|0.616|0.610|0.632|0.619|0.575|0.650|0.673|0.633|0.580|0.579|0.609|0.589|
> ||FDGPT|0.571|0.635|0.743|0.650|0.655|0.735|0.808|0.733|0.603|0.000|0.918|0.507|
> ||BARTScore|0.767|0.706|0.566|0.680|0.724|0.754|0.685|0.721|0.708|0.733|0.674|0.705|
> |**Story**|RoBERTa|0.588|0.586|0.660|0.611|0.540|0.504|0.539|0.527|0.571|0.569|0.657|0.599|
> ||RADAR|0.597|0.614|0.510|0.574|0.507|0.756|0.827|0.697|0.560|0.513|0.619|0.564|
> ||RAIDAR|0.860|0.837|0.851|0.849|0.757|0.799|0.735|0.764|0.814|0.830|0.889|0.844|
> ||ImBD|*0.949*|*0.904*|*0.973*|*0.942*|*0.984*|*0.989*|*0.974*|*0.983*|*0.973*|*0.986*|*0.996*|*0.985*|
> ||Ours|**0.998**|**0.959**|**0.990**|**0.982**|**0.997**|**0.999**|**0.977**|**0.991**|**0.990**|**0.999**|**0.999**|**0.996**|
> ||Ours(old)|0.999|0.954|0.996|0.983|0.990|1.000|0.981|0.990|0.987|0.999|0.999|0.995|
>
> Note: The RAIDAR results in **Table H** differ from those in Table 2 of the main text because we previously reported the AUC on the training set by mistake. This has been corrected in **Table H**.

---

> ### Author Response · Authors · 2025-11-20
>
> **Table I**. Use `mistralai/Mistral-7B-Instruct-v0.3` as the rewriting and scoring model for rewrite-based and logits-based model. The target model is GPT-3-Turbo. Comparison of different models across various datasets. The best performance is marked in **bold** and the second-best in *italic*.
>
> |Dataset|Likelihood|LRR|IDE|BARTScore|FDGPT|Binoculars|RoBERTa|RADAR|RAIDAR|ImBD|Ours|
> |:-|:-|:-|:-|:-|:-|:-|:-|:-|:-|:-|:-|
> |AcademicResearch|0.6378|0.6766|0.5222|0.5545|0.5779|0.6608|0.5156|0.7198|*0.8640*|0.6654|**0.9701**|
> |ArtCulture|0.5430|0.5602|0.5403|0.5467|0.5806|0.5106|0.6067|0.6146|**0.8640**|0.5101|*0.7936*|
> |Business|0.5171|0.5001|0.5992|0.5736|0.6349|0.5167|0.5599|0.5814|*0.7895*|0.7768|**0.9018**|
> |Code|0.6187|0.5664|0.5624|0.5697|0.5231|0.6376|0.5262|0.7017|*0.7858*|0.7612|**0.9351**|
> |EducationMaterial|0.6299|0.6208|0.5962|0.7440|0.5783|0.8022|0.6887|0.8543|0.8640|**0.9802**|*0.9014*|
> |Entertainment|0.6607|0.5842|0.5998|0.7230|0.5320|0.6970|0.7588|*0.8911*|0.8640|0.8217|**0.9945**|
> |Environmental|0.6717|0.5229|0.6132|0.6909|0.7720|0.7142|0.6820|0.6495|0.7895|*0.8204*|**0.9790**|
> |Finance|0.5682|0.5266|0.5462|0.6532|0.5232|0.6155|0.6746|0.6516|0.7895|*0.8547*|**0.8934**|
> |FoodCusine|0.5870|0.5184|0.5014|0.5971|0.6098|0.5225|0.5625|0.5209|*0.8640*|0.5267|**0.9751**|
> |GovernmentPublic|0.5217|0.5443|0.5867|0.6064|0.6747|0.5649|0.6128|0.6398|*0.8640*|0.5504|**0.8764**|
> |LegalDocument|0.7580|0.5343|0.7235|0.6521|0.5486|0.6318|0.5972|0.8143|0.7895|*0.9552*|**0.9699**|
> |LiteratureCreativeWriting|0.5898|0.6307|0.5792|0.6698|0.7064|0.5236|0.6183|0.8391|0.7895|*0.9623*|**0.9944**|
> |MedicalText|0.5063|0.5051|0.5387|0.5322|0.5699|0.5564|0.5227|0.6252|**0.8640**|0.5020|*0.8036*|
> |NewsArticle|0.6876|0.7281|0.5592|0.5159|0.5105|0.6834|0.6267|0.8554|*0.8640*|0.7703|**0.9755**|
> |OnlineContent|0.5571|0.5926|0.5633|0.5286|0.5088|0.6155|0.5956|0.6128|*0.7895*|0.7133|**0.9633**|
> |PersonalCommunication|0.5334|0.5997|0.5252|0.5018|0.5637|0.5204|0.5208|0.5855|*0.7895*|0.7164|**0.9177**|
> |ProductReview|0.6558|0.5341|0.5427|0.6447|0.6465|0.5507|0.6150|0.5918|*0.8640*|0.5605|**0.9672**|
> |Religious|0.7842|0.6537|0.8678|0.7092|0.5724|0.5801|0.5647|0.8682|0.7895|*0.8921*|**0.9223**|
> |Sports|0.5201|0.5215|0.5311|0.5668|0.6232|0.5209|0.5542|0.6032|*0.7895*|0.7342|**0.8494**|
> |TechnicalWriting|0.6431|0.6577|0.5027|0.5372|0.6232|0.7059|0.5188|0.7398|0.7895|*0.7963*|**0.9864**|
> |TravelTourism|0.6341|0.6689|0.5001|0.5387|0.6402|0.7258|0.5327|0.7411|*0.8640*|0.7053|**0.9724**|
> |Average|0.6107|0.5832|0.5762|0.6027|0.5962|0.6122|0.5931|0.7001|*0.8248*|0.7417|**0.9306**|
> |Std|0.076|0.064|0.081|0.072|0.066|0.083|0.065|0.111|0.037|0.145|0.059|

---

> ### Author Response · Authors · 2025-11-20
>
> **Table J**. Use `Qwen/Qwen2.5-7B-Instruct` as the rewriting and scoring model for rewrite-based and logits-based model. The target model is GPT-4o. Comparison of different models across various datasets. The best performance is marked in **bold** and the second-best in *italic*.
>
> |Dataset|Likelihood|LRR|IDE|BARTScore|FDGPT|Binoculars|RoBERTa|RADAR|RAIDAR|ImBD|Ours|
> |:-|:-|:-|:-|:-|:-|:-|:-|:-|:-|:-|:-|
> |AcademicResearch|0.5603|0.6113|0.5274|0.5294|0.6138|0.6226|0.5163|0.6374|0.7565|*0.9537*|**0.9901**|
> |ArtCulture|0.5701|0.5694|0.5755|0.5788|0.5436|0.5093|0.5701|0.5597|*0.7565*|0.7318|**0.8167**|
> |Business|0.5491|0.5069|0.5565|0.5615|0.5136|0.5317|0.5121|0.5403|0.7662|*0.8909*|**0.9116**|
> |Code|0.5676|0.6610|0.5825|0.5636|0.6590|0.6711|0.5890|0.5536|*0.7713*|0.6086|**0.9377**|
> |EducationMaterial|0.6277|0.6954|0.5584|0.8432|0.8208|0.8844|0.7243|0.7458|0.7565|**0.9910**|*0.9371*|
> |Entertainment|0.6417|0.5397|0.6216|0.6855|0.6527|0.6297|0.6675|0.7928|0.7565|*0.9851*|**0.9966**|
> |Environmental|0.7026|0.5457|0.6141|0.7008|0.5893|0.5701|0.6216|0.5711|0.7662|*0.9476*|**0.9910**|
> |Finance|0.6459|0.5567|0.5383|0.6700|0.5878|0.5743|0.6119|0.5731|0.7662|*0.9298*|**0.9634**|
> |FoodCusine|0.6035|0.5421|0.5263|0.6040|0.6140|0.6323|0.5584|0.5073|0.7565|*0.9221*|**0.9866**|
> |GovernmentPublic|0.5536|0.5479|0.5935|0.5385|0.6127|0.6240|0.5696|0.5787|0.7565|*0.9029*|**0.9350**|
> |LegalDocument|0.5964|0.5605|0.6415|0.5364|0.6407|0.6463|0.5281|0.5469|0.7662|*0.8494*|**0.8773**|
> |LiteratureCreativeWriting|0.6829|0.5142|0.5725|0.6821|0.6690|0.6158|0.5242|0.6860|0.7662|*0.9589*|**0.9884**|
> |MedicalText|0.5142|0.5332|0.5476|0.5105|0.5226|0.5417|0.5287|0.5640|0.7565|*0.7742*|**0.8581**|
> |NewsArticle|0.6227|0.6411|0.5556|0.5522|0.5835|0.6107|0.5150|0.7841|0.7565|*0.9398*|**0.9938**|
> |OnlineContent|0.5766|0.5170|0.5363|0.5689|0.5390|0.5204|0.5774|0.5736|*0.7662*|0.7252|**0.9370**|
> |PersonalCommunication|0.5165|0.5218|0.0000|0.5005|0.5241|0.5409|0.5115|0.5178|*0.7662*|0.6635|**0.8478**|
> |ProductReview|0.6575|0.5627|0.5021|0.6219|0.6144|0.6250|0.5835|0.5440|0.7565|*0.9227*|**0.9853**|
> |Religious|0.7143|0.6297|*0.8447*|0.6851|0.6267|0.6090|0.5846|0.7628|0.7662|0.6536|**0.9688**|
> |Sports|0.5496|0.5396|0.5194|0.5895|0.5250|0.5184|0.5065|0.5559|0.7662|*0.7972*|**0.8668**|
> |TechnicalWriting|0.5667|0.6068|0.5258|0.5583|0.6234|0.6448|0.5596|0.6312|0.7662|*0.9272*|**0.9902**|
> |TravelTourism|0.5667|0.6068|0.5258|0.5583|0.6234|0.6448|0.5596|0.6312|0.7565|*0.9662*|**0.9897**|
> |Average|0.5993|0.5719|0.5460|0.6019|0.6047|0.6080|0.5676|0.6123|0.7618|*0.8591*|**0.9414**|
> |Std|0.056|0.051|0.141|0.081|0.068|0.078|0.054|0.088|0.005|0.117|0.055|
>
> **Table K**. The average AUC scores of various detectors ($\pm$ standard deviations) under the experiments with different prompts. The best performance is marked in **bold** and the second-best in *italic*.
>
> ||News|Wiki|Story|
> |:---|:---|:---|:---|
> |Likelihood|0.606$\pm$0.052|0.546$\pm$0.041|0.630$\pm$0.102|
> |LRR|0.627$\pm$0.042|0.555$\pm$0.057|0.603$\pm$0.077|
> |Binoculars|0.596$\pm$0.059|0.708$\pm$0.092|0.735$\pm$0.101|
> |IDE|0.669$\pm$0.055|0.632$\pm$0.052|0.614$\pm$0.032|
> |FDGPT|0.582$\pm$0.059|0.694$\pm$0.098|0.708$\pm$0.108|
> |BARTScore|0.605$\pm$0.062|0.625$\pm$0.086|0.702$\pm$0.056|
> |RoBERTa|0.540$\pm$0.024|0.655$\pm$0.085|0.579$\pm$0.049|
> |RADAR|0.865$\pm$0.083|0.638$\pm$0.138|0.611$\pm$0.106|
> |RAIDAR|0.894$\pm$0.027|0.899$\pm$0.037|0.819$\pm$0.046|
> |ImBD|*0.970*$\pm$0.026|*0.954*$\pm$0.031|*0.970*$\pm$0.026|
> |Ours|**0.999**$\pm$0.001|**0.983**$\pm$0.006|**0.990**$\pm$0.013|

---

### Official Review · Reviewer_7oJT · 2025-10-31

**Soundness:** 3
**Presentation:** 3
**Contribution:** 2
**Rating:** 6
**Confidence:** 4

**Summary:**

This paper proposes a fine-tuning approach for pre-trained LLMs that aims to maximize the divergence in reconstruction errors between human-rewritten and LLM-rewritten text. The method is grounded in the assumption that LLM-generated texts form a subset of human-written texts, and that this relationship persists in their semantic projections—thus justifying the adopted distance-learning strategy. Evaluated through extensive experiments, the method is shown to outperform existing baselines. The primary contribution of this work is a novel perspective on modeling the divergence between LLM-generated and human-written text, coupled with comprehensive empirical validation of the method's effectiveness.

**Strengths:**

1. The paper presents a well-structured categorization of existing methods for detecting human-written and LLM-generated text, help building a clear roadmap and contextualizing its own research direction.

2. It introduces a novel perspective centered on maximizing the discrepancy between human-written and machine-generated text, which is supported by well-motivated theoretical assumptions.

3. The method's effectiveness is validated through extensive experiments, which include comparisons against a wide range of baselines across multiple datasets.

**Weaknesses:**

1. The paper mentions that "adaptively learn a distance function" to enhance detection performance, but the implementation of this adaptive learning process is not clearly described.

2. The pre-trained LLMs used in the experiments are not explicitly identified, making it difficult to assess the experimental setup with confidence. Given the importance of model selection in evaluating the contribution of the work, such implementation details should be unambiguously stated.

3. There is an apparent discrepancy between the reported "relative gain" (e.g., claims of 100% improvement) and the modest absolute improvements observed in the AUC scores. Although the calculation method is provided in the appendix, the presentation in the main text can be misleading. Highlighting large relative gains derived from small absolute baselines may overstate the practical advancement and requires clearer contextualization to avoid confusion.

**Questions:**

My questions and comments have been stated in the detailed comments.

---

> ### Author Response · Authors · 2025-11-19
>
> > Response A1: Implementation of adaptive learning  (W1).
>
> We first provide an outline of our adaptive learning algorithm. We next detail the implementation. To enhance reproducibility, we will make our complete workflow publicly available on GitHub/Hugging Face.
>
> First, our adaptive learning algorithm can be summarized into the following three steps:
> 1. Collect a dataset of human-authored text (denoted by $\mathcal{D}_h$) and prompt the target LLM (e.g., GPT-4o) to obtain an LLM-generated dataset (denoted by $\mathcal{D}_m$).
>
> 2. For each text $X\in \mathcal{D}_h\cup \mathcal{D}_m$, prompt an open-source lightweight LLM (specified below) to rewrite it $K$ times, and denoted the $K$ reconstructions by $\widetilde{X}_1,\cdots,\widetilde{X}_K$.
>
> 3. Learn a distance function $d_\phi$ that maximizes the difference in reconstruction errors between $\mathcal{D}_h$ and $\mathcal{D}_m$:
>
> $$\max _{\phi} \mathbb{E} _{X \sim D _{h}} \bigg[ \frac{1}{K} \sum _{k=1}^{K} d _{\phi}(X, \tilde{X} _{k}) \bigg] - \mathbb{E} _{X \sim D _{m}} \bigg[ \frac{1}{K}\sum _{k=1}^K d _\phi(X, \tilde{X} _k) \bigg],$$
>
> where $d_\phi(X_1, X_2) = |  \log p_\phi(X_1)/|X_1| - \log p_\phi(X_2)/|X_2| |$ and $p_\phi$ is a language model whose architecture will be detailed below.
>
> Next, we detail our implementation. We use `google/gemma-2-9b-it` as the rewriting model, with $K=4$ in our experiments. For simplicity, the model $p_{\phi}$ in the distance function adopts the same architecture, `google/gemma-2-9b-it`. We train this distance function with a learning rate of $10^{-4}$, for 2 epochs, using a batch size of 1. The optimizer is AdamW combined with a CosineAnnealingLR scheduler. The size of the training dataset depends on the experimental setting: when more than 1000 training texts are available, we randomly select 1000 samples for training; otherwise, we use the maximum available number of training texts, typically fewer than 600 samples. Our fine-tuning hyperparameters were reported in Appendix C of our submission: We fine-tune the model using LoRA, implemented via the peft library, with rank $=$ 8, LoRA alpha $=$ 32, and LoRA dropout $=$ 0.1; all other parameters follow default settings.
>
> Please let us know if you have any additional questions about the learning process.
>
> > Response A2: Pre-trained LLMs used in the experiments (W2)
>
> As disclosed in our previous comment, our algorithm involves two pre-trained LLMs: one for rewriting and the other for computing the distance function. Both were set to `google/gemma-2-9b-it`, and the parameter $\phi$ in the second model was trained starting from its pre-trained weights.
>
> > Response A3: Use of relative gain (W3)
>
> We clarify that this metric has been employed in prior work [1] to report improvements of a proposed detector over a baseline. Our use of this metric was intended to maintain consistency with existing literature, not to exaggerate our conclusions.
>
> To address your comment, we will report both  absolute AUC gains and relative improvements (with the exact formula provided in the main text) to ensure that our results are presented clearly and transparently, if our paper is accepted.
>
> **Reference**
>
> [1] Bao, G., Zhao, Y., Teng, Z., Yang, L., \& Zhang, Y. (2023). Fast-detectgpt: Efficient zero-shot detection of machine-generated text via conditional probability curvature. arXiv preprint arXiv:2310.05130.

---

### Author Response · Authors · 2025-11-21

Dear reviewers,

We sincerely appreciate all reviewers' thoughtful comments. In response to these comments, we have conducted various experiments whose results are detailed in **Table A** to **Table K**, and revised our theories under less restrictive assumptions.

We noticed that similar comments were raised by different reviewers. For your convenience and easy navigation, we have bolded all key information (e.g., **Table B**, **Response C3**). You may use your browser's search function to quickly locate them.

We hope our responses address your concerns. Please feel free to ask any further questions.

Best regards,

---

### Author Response · Authors · 2025-12-03
**To AC**

Dear AC,

We sincerely thank the reviewers for their time, effort, and constructive feedback. For your convenience, we provide a concise summary of our responses and major changes in the revised paper (highlighted in red) to aid your decision making. Note that our responses are organized as Response A1, A2, A3, B1, etc., and you may use your browser’s search function to quickly locate them.

- **Novelty & Contributions (Reviewer J3u5):** We added a new theorem (**Proposition 3**) and related discussions to deeply justify and motivate our learned distance function. In particular, **Propositions 1 & 2** provide geometric insights that motivate our use of rewrite-based methods, while **Proposition 3** supports our choice of an adaptively learned distance for implementing these methods (see **Response D6** for further details). We also reframed the paper to more accurately reflect our theoretical contributions.
- **Assumptions (Reviewers oNqV, pN4E):** We revised our theory under less restrictive assumptions and verified it empirically (**Appendix A**, **Responses B1 & C1**). We keep the original version in the main text for its geometric intuition and interpretability, while presenting the more rigorous, newly developed theory in the appendix. This addresses concerns while keeping minimal changes to the main paper (**Response C5**).
- **Experiments & Runtime (Reviewers oNqV, pN4E, J3u5):** We ensured fairness in our experiments (**Section 4, Response C4**), demonstrated generalization across models/datasets (**Response C2**), investigated small declines to further improve the algorithm (**Response B2**), reported runtime excluding model-loading overhead (**Appendix B, Response D7**), and discussed scalability for long documents and large-scale batch detection (**Response C3**).
- **Clarifications & Presentation (Reviewers 7oJT, oNqV, J3u5):** We clarified implementation details (**Appendix B, Responses A1 & A2**) and reported both absolute and relative gains (**Tables 1-2, B1-B3** in the revised paper & **Response A3**).

---

### Meta-Review · Area_Chair_zqm4 · 2025-12-17

**Summary:**

The major concerns of the reviewers are about the assumptions and the fairness of experimental comparison. Reviewers pointed out that the geometric assumptions underlying the theory may be unrealistic and questioned if the proposed approach relies on access to the target LLM. Reviewers also pointed out that the proposed method requires training, while the key baselines are zero-shot and training-free, making the comparison unfair. The additional computational overhead of the approach was identified as a potential practical limitation.

**Reviewer Concerns:**

The concerns about theory assumptions, the experimental setting (if the method needs access to target LLM) and lack of implementation details were addressed. Some reviewers still have concerns about the novelty and additional computational overhead.

**Reviewer Scores:**

Reviewer 7oJT and oNqV are likely to keep the original positive scores. Reviewer J3u5 and pN4E are likely to keep the decisions based on the discussion.

---

### Decision · Program_Chairs · 2026-01-26

Accept (Poster)